

# Non–linear intensification of Sahel rainfall as a dynamic response to future warming

Jacob Schewe[1] and Anders Levermann[1,2,3]

[1]Potsdam Institute for Climate Impact Research, Potsdam, Germany
[2]Institute of Physics, Potsdam University, Potsdam, Germany
[3]Lamont-Doherty Earth Observatory, Columbia University, New York, USA

*Correspondence to:* Jacob Schewe (jacob.schewe@pik-potsdam.de)

**Abstract.** Projections of the response of Sahel rainfall to future global warming diverge significantly. Meanwhile, paleoclimatic records suggest that Sahel rainfall is capable of abrupt transitions in response to gradual forcing. Here we present climate modeling evidence for the possibility of an abrupt intensification of Sahel rainfall under future climate change. Analyzing 30 coupled global climate model simulations, we identify seven models where central Sahel rainfall increases by 40% to 300% over the 21st century, owing to a northward expansion of the West African monsoon domain. Rainfall in these models is non–linearly related to sea surface temperature (SST) in the tropical Atlantic moisture source region, intensifying abruptly beyond a certain SST warming level. We argue that this behaviour is consistent with a self–amplifying dynamic–thermodynamical feedback, implying that the gradual increase in oceanic moisture availability under warming could trigger a sudden intensification of monsoon rainfall far inland of today's core monsoon region.

## 1 Introduction

The Sahel is a wide semi–arid belt spanning the African continent south of the Sahara desert, and is home to a large population strongly reliant on agriculture (Sissoko et al., 2010). Its climate has been characterized by devastating droughts, such as in the 1970s and 80s (Folland et al., 1986; Zeng, 2003), alternating with episodes of abundant rainfall, and even destructive rain and flood events (Tschakert et al., 2010; Tarhule, 2005). The 1970s/80s drought, which resulted in persistent food shortage and widespread famine (Nicholson, 2013), has been attributed to anthropogenic reflective aerosols as well as variations in Atlantic sea surface temperature (SST), which may have been partly human–induced and partly due to natural variability (Giannini et al., 2003; Biasutti and Giannini, 2006). Rainfall has partially recovered more recently (Lebel et al., 2009), a trend that has been attributed both to the direct radiative effect of anthropogenic greenhouse gases (Dong and Sutton, 2015) and to SST warming, especially in the Mediterranean (Park et al., 2016).

While coupled climate models generally capture the temporal pattern of the 1970s/80s drought (Biasutti and Giannini, 2006), most simulations from the recent Coupled Model Intercomparison Project phase 5 (CMIP5) (Taylor et al., 2012) underestimate its magnitude Biasutti (2013). At the same time, projections of future rainfall changes diverge substantially across the models (Power et al., 2012). The CMIP5 multi–model mean was shown (Biasutti, 2013) to exhibit only a slight increase in overall Sahel rainfall, with a wetting trend over the central and eastern Sahel and drying over the westernmost part, under the highest



Representative Concentration Pathway (van Vuuren et al., 2011), RCP8.5. Individually, some of the models project a much stronger rainfall increase, while others even project an overall decrease.

This uncertainty in future projections raises questions about potential mechanisms of change that may be present in some models but not others, and that may be responsible for the large differences between models. In particular, paleoclimatic records
suggest that Sahel rainfall is capable of abrupt transitions in response to gradual forcing (DeMenocal et al., 2000; McGee et al., 2013); and theoretical studies have demonstrated that such a non–linear response can in principle arise from internal monsoon dynamics (Levermann et al., 2009; Seshadri, 2016). In this study, we examine Sahel rainfall in state–of–the art climate model simulations and show that in those models that exhibit the strongest rainfall increase, this increase is non–linearly related to the SST warming in the tropical Atlantic moisture source region. We argue that this behaviour is consistent with the theory and
paleoclimatic evidence mentioned above. Considering that this non–linear rainfall response may be more pronounced in some models than in others may contribute to understanding the differences between the models' future projections.

## 2   Methods and Results

We investigated Sahel rainfall in 30 Coupled Model Intercomparison Project phase 5 (CMIP5) (Taylor et al., 2012) global climate models under RCP8.5 (see Appendix A). Based on projected summer (July–September) rainfall increase across the
15 central and eastern Sahel (0-30°E, 10–20°N), we singled out the seven "wettest" models (hereafter referred to as Wet7), with a doubling of average summer rainfall by the end of the 21$^{st}$ century in the mean across these models (Fig. 1). In contrast, the mean over the 23 other models in the ensemble exhibits only a weak wetting trend of less than 20%, due to weak trends in the individual models and also to the fact that some models show a drying trend. The positive trend that has been found in the CMIP5 ensemble as a whole is therefore largely due to the Wet7 subset (cf. Roehrig et al., 2013). At the same time, we note
that the Wet7 models perform better than average in reproducing the magnitude of the 1970–1989 drought period, the three MIROC models especially being very close to observations (Fig. 1 and 2).

The seasonal distribution of the rainfall change in the Wet7 shows a clear monsoonal shape (Fig. 3). Generally, the rainfall increase occurs over a broad region between 10 and 20°N, i.e., extending into today's Sahara desert (Fig. 4). Conversely, rainfall decreases somewhat in the more humid regions around the Gulf of Guinea and the West coast. This pattern corresponds to an
25 inland shift compared to the present–day rainfall regime. At the same time, the near–surface, southwesterly winds intensify in the northern and eastern parts of the Sahel, near the positive rainfall anomaly, while they do not change much near the coast (Fig. 5). This suggests that the rainfall increase is not simply a consequence of thermodynamic changes, but part of a shift in West African monsoon circulation dynamics.

Sahel rainfall has been linked to Atlantic ocean SSTs via evaporation rate and moisture supply (e.g., ref. (Giannini et al.,
2003)). In order to examine temporal patterns of rainfall and SST change more closely, we average each model's summer rainfall over a subregion of the Sahel (solid boxes in Fig. 4) in which the rainfall increase is particularly pronounced, and which is more inland than the present–day core monsoon region. Similarly, we identify for each model a region in the tropical Atlantic ocean (dashed boxes in Fig. 4) as the main source of additional moisture influx into the Sahel, based on the lower–troposphere



moisture flux anomalies (arrows in Fig. 4). A substantial part of today's Sahel moisture is sourced from the Mediterranean sea (Fontaine, 2003; Rowell, 2003), and the Wet7 models indeed show some intensification in moisture influx from the north, but the increase in Atlantic moisture influx is larger.

Sahel rainfall generally increases as the surface of the oceanic moisture source region warms (Fig. 6). But this relation is

not linear. Rainfall shows little response to SST changes within a range of approx. 1°C around the present–day value; but when SST increases beyond this point, rainfall shifts abruptly to a stronger level, where it then keeps increasing as SST rises further. Given the convex shape of the temperature forcing over time, the abruptness of the rainfall response is expected to be less apparent in the time domain, where rainfall appears relatively stable over the historical period, before it begins increasing strongly in the 21st century (Fig. 7).

Numerous paleoclimatic reconstructions reveal abrupt shifts in monsoon systems in Asia (Gupta et al., 2003; Wang et al., 2008) and Africa (DeMenocal et al., 2000; McGee et al., 2013; Weldeab et al., 2007) before and throughout the Holocene. In those cases, external forcing through changes in solar insolation was much more gradual than that associated with modern anthropogenic climate change. A physical mechanism has been proposed to explain such abrupt shifts in large–scale monsoon rainfall in response to gradual forcing (Levermann et al., 2009, 2016): While the summer monsoon circulation is initiated

by differential warming of land and ocean in spring, it is latent heat release from precipitation that maintains the land–sea atmospheric temperature contrast throughout the summer and thus drives the monsoon winds into the continental interior. The monsoon winds in turn supply the moisture necessary to maintain precipitation. Summer monsoon rainfall is thus powered by a positive feedback between moisture inflow and atmospheric heating. This positive moisture–advection feedback gives rise to a threshold behaviour with respect to external quantities that govern the energy budget of the monsoon; in particular, in

this simplified theory, there is a minimum atmospheric humidity in the oceanic moisture source region below which such a monsoon circulation cannot be maintained (Schewe et al., 2012).

This framework has been used to explain abrupt variations in monsoon strength documented in Asian speleothem (Schewe et al., 2012) and pollen records (Herzschuh et al., 2014), but has not yet been applied to modern monsoon systems. We suggest that it is also useful for understanding the projected Sahel rainfall changes in the Wet7 models. Today, the West African

monsoon is most active between the Gulf of Guinea coast and the southern edge of the Sahel (Nicholson, 2013). Rainfall declines towards the continental interior, and while central and eastern Sahel rainfall still exhibits a clear seasonality, it is relatively weak and erratic (compared to e.g. the Indian monsoon with its intense rainfall throughout much of the subcontinent). An increase in evaporation due to ocean warming in the tropical Atlantic increases moisture availability. Once atmospheric humidity exceeds the monsoon threshold even in the more continental parts of the Sahel, the moisture–advection feedback can

amplify the monsoon response by enhancing monsoon winds and moisture influx. In this case, these inland regions become increasingly connected with the oceanic moisture source, and benefit from further increases in oceanic evaporation. This framework can explain the observed shape of the rainfall response in both time and SST domain (Fig. 8).



## 3  Discussion and Conclusions

This explanation of an abrupt intensification of inland monsoon rainfall in the Sahel region is consistent with studies suggesting a substantially wetter Sahel, and Sahara, region in past climates compared to today (DeMenocal et al., 2000; Gasse, 2000). It is also consistent with theories linking rainfall changes in the Sahel to a combination of a local (through radiative forcing

changes) and a remote (through tropical SST impacts on atmospheric stability) forcing mechanism (Giannini, 2010; Giannini et al., 2013; Seth et al., 2010). In a warming world, the remote mechanism would increase atmospheric stability especially in places with oceanic influence, and make it harder for convection to set in. Acting in the other direction, the local mechanism would directly warm the surface and decrease vertical stability over land. The mechanism we suggest here would act on top of these two mechanisms, and help explain the abruptness of the Sahel rainfall response to global warming. It would particularly

affect the more continental parts of the region. We note that part of the increased moisture influx is through westerly winds near 10°N, a flow called the West African Westerly Jet (Pu and Cook, 2010, 2012). While its intraseasonal dynamics are somewhat distinct from the more southerly monsoon flow across the Gulf of Guinea, on a seasonal timescale both are driven by the pressure—and thus, temperature—gradient between the eastern Atlantic and the Sahel, and would be subject to the dynamical feedback mechanism described above. Consideration of this mechanism may help to make sense of the diversity of model

projections, and eventually establish a more consistent understanding of the Sahel's future climate in a warming world.

## Appendix A:  Models and data

We analyzed simulations from the BNU-ESM, CanESM2, FGOALS-g2, MIROC-ESM-CHEM, MIROC5, MIROC-ESM, NorESM1-M (Wet7 subset), ACCESS1-0, ACCESS1-3, CESM1-BGC, CESM1-CAM5, CMCC-CM, CMCC-CMS, CMCC-CESM, CNRM-CM5, EC-EARTH, FIO-ESM, GFDL-CM3, GFDL-ESM2M, GFDL-ESM2G, GISS-E2-H, GISS-E2-R, HadGEM2-

ES, HadGEM2-CC, inmcm4, IPSL-CM5A-LR, IPSL-CM5A-MR, IPSL-CM5B-LR, MRI-CGCM3, and MPI-ESM-MR global climate models, driven by historical forcing and the RCP8.5 greenhouse gas concentration scenario (Meinshausen et al., 2011). Simulation data was obtained from the CMIP5 archive at http://cmip-pcmdi.llnl.gov/cmip5/. Where several realizations of the same model simulation were available, we used the r1i1p1 configuration, since this one was available from all models. Near–surface wind data was not available for FGOALS-g2, therefore we show 850mb wind. CRU TS3.1 monthly precipitation data

was obtained from http://badc.nerc.ac.uk.

*Author contributions.*  J. Schewe and A. Levermann designed the research. J. Schewe carried out the analysis and wrote the paper, with contributions from A. Levermann.

*Competing interests.*  The authors declare that they have no conflict of interest.



*Acknowledgements.* We acknowledge the World Climate Research Programme's Working Group on Coupled Modelling, which is responsible for CMIP, and we thank the climate modeling groups for producing and making available their model output. J.S. received funding through the Leibniz society's EXPACT project.



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



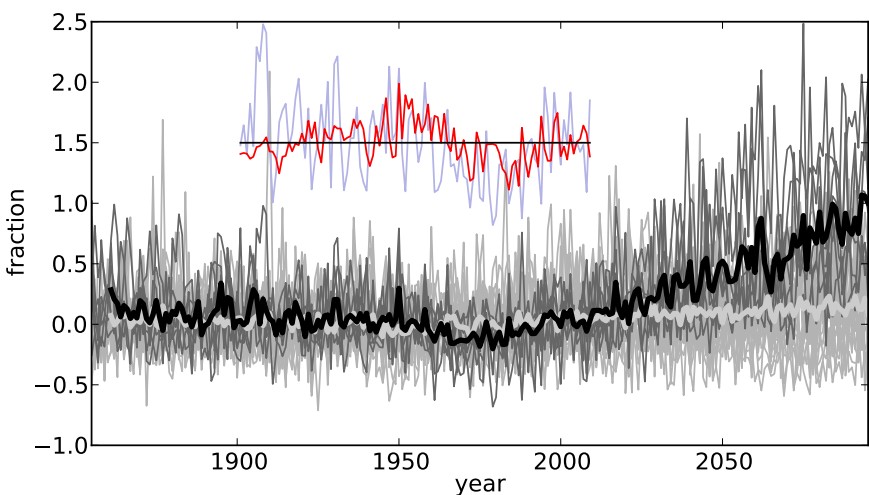

**Figure 1.** Sahel summer rainfall in models and observations. Grey lines show Central and Eastern Sahel (0-30°E, 10–20°N) July–September rainfall in the subset of seven GCMs investigated in this paper (dark) and in 23 other CMIP5 GCMs (light) under historical forcing and the RCP8.5 greenhouse gas concentration scenario. Shown is the deviation from the 1900–1999 average, as fraction of that average. The thick lines indicate the averages of each set of models. The inset shows the CRU TS3.1 observational data set (Harris et al., 2014) covering 1901–2009 (red) and the corresponding portion of the MIROC–ESM–CHEM simulation (light blue), in the same units as the other data but offset by 1.5 in the vertical for clarity.





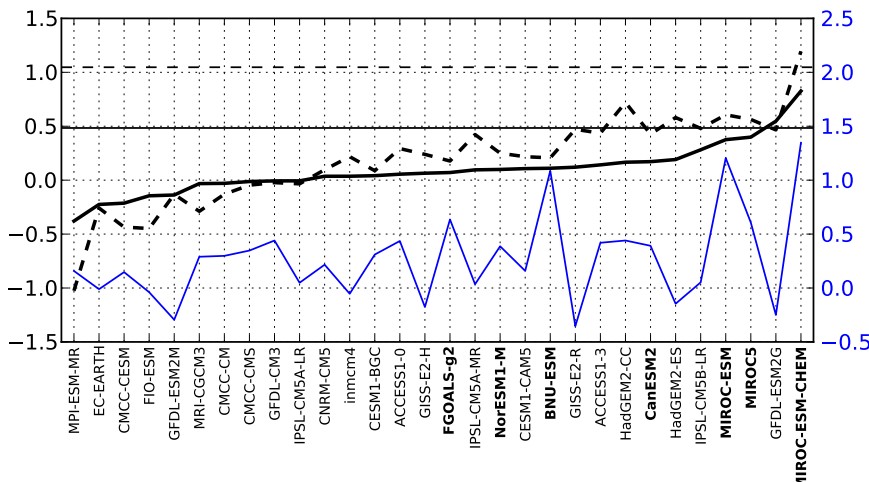

**Figure 2.** Magnitude of recent Sahel drought in models and observations. Solid black line shows the difference in average summer precipitation (mm/day, averaged over 0-30°E, 10-20°N) between the 1970–1989 drought period and the rest of the observational period ("non–drought", 1901–1969 and 1990–2009). Dashed black line shows the same difference divided by the standard deviation of the "non–drought" period. Horizontal lines show the respective observed value. Blue line shows the rainfall change by the end of the century (2071–2100) compared to the 1901–1999 average, as fraction of that average. Models that belong to the Wet7 subset are marked in bold.





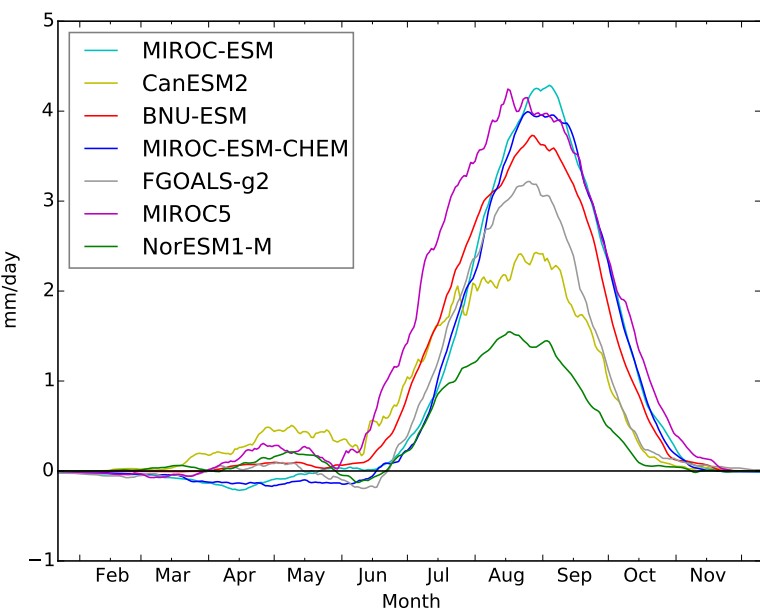

**Figure 3.** Difference in average Sahel daily precipitation between the end of the 20[th] century (1970–1999) and the end of the 21[st] century (2070–2099); all timeseries filtered with a 6–week running mean.



**Figure 4.** Simulated changes in Sahel summer climate under RCP8.5 in the Wet7 models. For each model the differences in July–September rainfall (colours), sea surface temperature (SST, greyscale; contour spacing is 0.5K), and moisture flux integrated vertically over the three bottom–most pressure levels (1000, 925, and 850 mb; arrows) are shown between the 20th century (1900–1999) and the end of the 21st century (2070–2099). Solid (dashed) boxes show the regions over which rainfall (SST) differences are averaged for Fig. 6. The color scale, vector scale, and coordinate labels of the top panel apply to all panels.



**Figure 5.** As Fig. 4, but arrows show changes in near–surface winds (850mb winds for FGOALS-g2, where near–surface wind speed was not available).





**Figure 6.** Median Sahel July–September rainfall for different intervals of SST change (interval width 0.25°C). Bars illustrate the deviation from the 1900–1999 rainfall average (horizontal black line), and SST change is also relative to the 1900–1999 average. Bars are only shown if at least 5 years fall into the respective temperature interval.





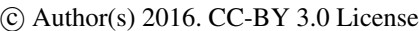

**Figure 7.** Mean Sahel July–September rainfall in 15-year intervals. Bars illustrate the deviation from the 1900–1999 rainfall average (horizontal black line).




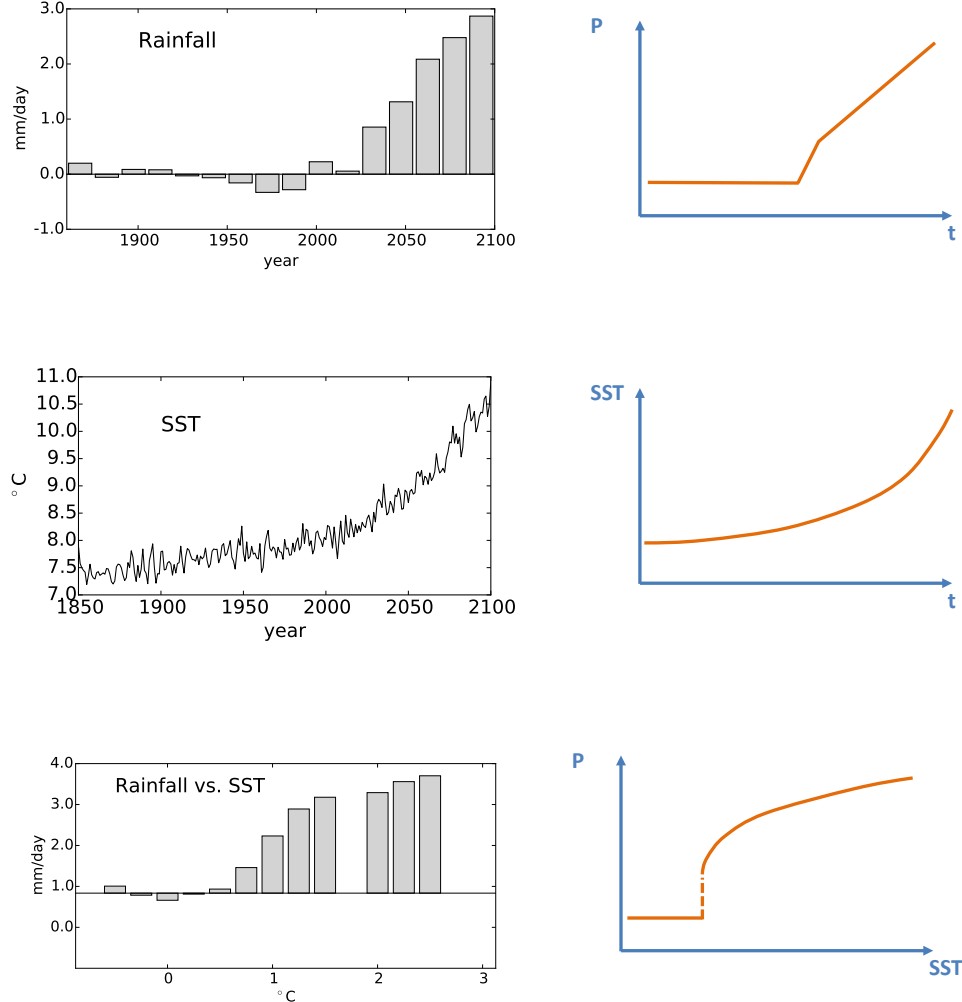

**Figure 8.** Comparison of model simulations with the concept of a monsoon threshold. Panels on the left show rainfall change over time (top), SST over time (middle), and rainfall change over SST change (bottom), from the MIROC-ESM-CHEM simulation, for the regions indicated in Fig. 2. Panels on the right show illustrative functional forms that qualitatively match those of the simulation data. The functional form of the rainfall evolution (P, top) can result from the combination of a convex sea surface temperature evolution (SST, middle) and a concave P–SST relationship with a moisture availability threshold (bottom); such a P–SST relationship arises analytically from the moisture–advection feedback (Schewe et al., 2012). The top left and bottom left panels are equivalent to the top panels in Fig. S4 (Supplementary) and Fig. 3, respectively.