# Peer review of "Non–linear intensification of Sahel rainfall as a dynamic response to future warming"

_Earth System Dynamics, 2016_

## Short Comment (SC1) · 3 Jan 2017

P.-A. Monerie

pmonerie@gmail.com

Page 1 - There is a typo, line 22 "(Biasutti, 2013) - line 24, the reference should be placed after "Sahel rainfall". You can also cite Fontaine et al. (2011), among others Fontaine B, Roucou P, Monerie P-A (2011) Changes in the African monsoon region at medium-term time horizon using 12 AR4 coupled models under the A1b emissions scenario. Atmos Sci Lett 12:83–88. doi:10.1002/asl.321 Page 2 - Line 18 "The positive trend.." is in fact obtained in at least 80 % of the CMIP5 simulations in Biasutti (2013), it is not only due to the wet7 - Line 19-21: The Wet7 is able to reproduce the 1970-1989 drought magnitude, but what is your conclusion? Do you think these models projections to be more reliable? Do there is a link between the projection and a models

ability to reproduce the current climate? - Line 22: "The seasonal distribution.." It is also the case for the other models (not only with the Wet7). You do not comment the large spread obtained with Figure3. Page3 - Line 1-3: Is it in contradictory with Park et al. (2016)? - Line 4: Are you analysing the global SSTs? This sentence is not clear Did you found the same results focusing on the North Atlantic Ocean, or the Mediterranean Sea? Figure - Figure7: if we only consider the period with a contiuous increase in the GHGs concentration, (the RCP8.5 emission scenario starts in 2005-2006), is the precipitation increase so abrupt?

––––––––––––––––––––––––––––––––––––

---

## Short Comment (SC2) · 3 Jan 2017

Thanks for the comments.

We will include the references and try to answer as many of the questions as possible.

Best wishes, Anders
* * *

---

## Referee Comment (RC1) · M Gaetani (Referee) · 16 Jan 2017

General comments

In this paper, future projections of Sahel rainfall are analysed. The authors show evidence for the possibility of an abrupt intensification of rainfall under future climate change, proposing a mechanism based on the non-linear response of precipitation to tropical Atlantic warming. The addressed scientific question is fully relevant to the scope of ESD.

The paper is written in a clear and concise manner. Wording is precise and the organisation of the manuscript facilitate the comprehension of background, motivation,

method, results and conclusions. Title and abstract effectively drive the reader into the focus of the paper. Motivation and objective are clearly stated, data and method are appropriate to address the scientific questions. Data used are open access and the method is straightforward and clearly described, making easy the reproduction of the study. Results are robust and correctly interpreted, supporting substantial conclusions, which are precisely outlined. Essential existing literature is cited to introduce the background on the subject, to motivate the study and indicate the originality of the contribution, as well as to put results and conclusions in the context of the current knowledge on the subject.

These results represent a noticeable contribution to the debate on future projections of the Sahel rainfall, though some aspects should be improved (see specific comments below).

Specific comments

Page 2, line 19: Add Park et al. 2015, on the northern-hemispheric differential warming impact on the projected Sahel rainfall (http://www.nature.com/articles/ncomms6985).

Page 2, lines 19-21: Other than the magnitude of the big drought, it would be interesting also to analyse the model ability in reproducing the decadal variability in the historical period.

Page 2, lines 31-33: What does "particularly pronounced" exactly mean? Please detail the method to select precipitation and moisture transport boxes in Figure 4.

Page 3, lines 1-3: This is the main issue in the paper. You state that "a substantial part of today's Sahel moisture is sourced from the Mediterranean", and this has been shown to be one of the key areas in future GW scenarios (Park et al. 2016). Then you state that the flux from the Mediterranean is negligible compared to the tropical Atlantic. This should be substantiated. A comparison between tropical Atlantic and Mediterranean moisture sources should be shown, as well as a comparison between

the effects on precipitation of the SST warming in both the basins.

Figure 2: Is the magnitude of drought computed as drought minus no-drought? So models reproducing drought should give negative values. I think it would be better to change the sign, also for coherence with the changes by the end of the 21st century. Moreover, in the text you state that Wet7 models are "better than average in reproducing drought magnitude", therefore I suggest to add the multimodel mean to the plot, to show this.

Figure 6: How do you obtain precip-SST plots? Do they refer to 21st century only? Please clarify this.

Figure 7: It would be very useful to add the SST time series to precipitation.

Figure 8: I think showing again MIROC-ESM-CHEM is redundant, better to show the multimodel mean, alongside the conceptual scheme.

---

## Short Comment (SC3) · 16 Jan 2017

Dear Dr. Gaetani,

thank you very much for your constructive review. We will address all your comments in a revised manuscript. A few initial responses for clarification:

On the Mediterranean: We did not want to imply that the moisture flux from the Mediterranean is negligible; only that the simulated \*increase\* in moisture flux from the Mediterranean is smaller than the increase in moisture flux from the Atlantic.

Figure 2: Indeed, positive values indicate that the drought is reproduced with correct sign. We will consider reversing the sign in this figure.

[Figure]

Figure 6: This refers to the whole period 1850-2100. Sorry for not being clear about this in the figure caption.

[Figure]

---

## Referee Comment (RC2) · Anonymous Referee #2 · 14 Feb 2017

Recommendation: Accepted after minor revisions

The present paper is concise, well written and contributes in a clear manner for the discussion of a relevant problem: what may drive rain anomalies in a future climate scenario. More precisely, the authors suggest that the SST anomaly during a warming future period in the African-Equatorial region will induce an enhancement of the moisture-advection feedback and an abrupt intensification of the African monsoon. The authors also argue that neither the local radiative land forcing nor the remote SST influence would be sufficient to explain the expected rain change. However some points must be clarified.

Major points

1 - Authors have chosen the wettest models (wet7 subset of models from the CMIP5 ensemble) in the Sahelian region by the end of 21th century. Any other criterion could be chosen (e.g. the driest models). There is no explicit evidence in the paper why the wet7 subset is the one from which one expects better future rain predictions in Sahel. Models have biases, both in the average, standard deviation, extremes etc. as we compare model simulations with a reference observed period (like that shown in Fig. 1: from ∼1900 to ∼2000).

Moreover, authors say in pg. 2 lines 19-21: 'At the same time, we note that the Wet7 models perform better than average in reproducing the magnitude of the 1970–1989 drought period'. I should stress that, from Fig. 1 you cannot conclude the above sentence from information displayed on Fig. 1.

Therefore, a simple table of statistical biases should be included to clarify the performance of wet7 subset in the reference period as compared with other models.

2 - The non-linear response P-SST (on Fig. 8), which is the main paper's result is quite interesting but deserves much explanation. Only a single phrase at the end of Section 2 refers Fig. 8 and the non-linear relationship. Authors should clarify some points.

3 - Fig. 8 shows the non-linear relationship P-SST. It is evident that some smoothing and composite averaging is done to minimize noise. Please clarify that. Do P and SST are taken over some running averages? Do results change if the binning length is shortened? What is the delay in the proposed moisture-advection feedback?

Minor points

4 - In pg. 1, line 13: Precise in the text the periods with episodes of heavy rainfall in the text.

5 - Fig. 1 In the caption, the thick grey curves are quite indicative of the trend. However, the light grey lines for the two sets of models are totally overlapped becoming useless. It would be much clear to show the temporal curves of the interval range, i.e. the

minimum and the maximum over each model set (the 7-model set and the 23-model set).

6 - pg. 2, lines 27-28 Authors say: 'This suggests that the rainfall increase is not simply a consequence of thermodynamic changes, but part of a shift in West African monsoon circulation dynamics' Justify the first sentence please.

7 - Fig 2. Caption is 'Solid black line shows the difference in average summer precipitation (mm/day, averaged over 0-30_E, 10-20_N) between the 1970–1989 drought period and the rest of the observational period ("non–drought", 1901–1969 and 1990–2009).'Therefore we expect a negative anomaly: 'drought period' minus 'observed non-drought period'. Like in other parts of the paper, it is not understood what Is the subtrahend and the minuend of the subtraction.

8 - Fig. 3. Caption: Change the word 'Difference' to the word 'Deviation from'. Difference between A and B is A minus B, so please clarify the caption.

9 - Fig. 4 There is no grey color bar for the SST anomalies. At least indicate where is the zero value.

---

## Short Comment (SC4) · 17 Feb 2017

Dear reviewer,

thank you very much for your constructive comments, which we will address in detail in our final response.

As a quick clarification: The grey shades in Figure 4 do not include the zero line, i.e. the sea surface gets warmer everywhere in this sector. Values shown range approximately from 1°C (light grey) to 5°C (dark grey). We will include a precise indication (color bar or marked contour) in the revised manuscript. In Figure 2, the black line shows "non-drought period" minus "drought period"; hence the positive values for most models.

[Figure]

The caption is indeed unclear on this, as was also noted by reviewer #1. I apologize for this, and will correct it.

---

## Author Response (AR1)

Final author comments

We would like to thank all reviewers for their helpful and constructive comments. Below we detail our responses to the individual comments.

Referee #1 (M. Gaetani)

*Page 2, line 19: Add Park et al. 2015, on the northern-hemispheric differential warming impact on the projected Sahel rainfall (http://www.nature.com/articles/ncomms6985).*

> Thank you for pointing out this paper which indeed is relevant in this context. We have added the reference.

*Page 2, lines 19-21: Other than the magnitude of the big drought, it would be interesting also to analyse the model ability in reproducing the decadal variability in the historical period.*

> We agree that the magnitude of the 70s/80s drought is only one of many aspects of historical model performance. Biasutti (2013) have studied the CMIP5 model ensemble's performance in some more detail and report that, similar to the CMIP3 ensemble, most models underestimate the multi-decadal oscillations observed in Sahel rainfall. At the same time, they state that "Individual coupled simulations do reproduce the decadal ups and downs of the observed Sahel rainfall: Held et al. [2005] documented the case of one GFDL model, and we see the same for one MIROC model (not shown)". So, while a more in-depth evaluation of the CMIP5 models' historical simulations is beyond the scope of our present study, we note that our results are consistent with those of Biasutti (2013) and that readers might refer to that study. In particular, the inset in our new Fig. 2 (previously Fig. 1) shows that the MIROC-ESM-CHEM model not only reproduces the magnitude of the 70s/80s drought but also the multi-decadal variability during the rest of the 20$^{th}$ century (despite an overestimation of inter-annual variability).

> We have amended the corresponding paragraph in the manuscript as follows: "Although we focus here on the future projections, we also note that the Wet7 models perform better than average in reproducing the magnitude of the 1970–1989 drought period, the three MIROC models especially being very close to observed values (orange lines in Fig. 1, and inset in Fig. 2). This observation is consistent with a more comprehensive analysis of the CMIP5 models for the historical period (Biasutti, 2013), which found that past multi–decadal variability is underestimated by all except a few models, one MIROC model among them. It may serve as an additional motivation to further study the future projections by these models, which we do in the following."

*Page 2, lines 31-33: What does "particularly pronounced" exactly mean? Please detail the method to select precipitation and moisture transport boxes in Figure 4.*

We have amended the text as follows: "In order to examine temporal patterns of rainfall and SST change more closely, we average each model's summer rainfall over a rectangular subregion of the Sahel (solid boxes in Fig. 4 and 5). The subregions are chosen to encompass an area where the rainfall increase is substantial in both absolute (Fig. 4) and relative terms (Fig. 5), and to be similar in size and location across the different models' grids (except for CanESM2 where the rainfall increase is located further east than in the other models). Thus, the subregions are generally located northward of the present–day core monsoon regions, which also see rainfall increases but less pronounced in relative terms."

We have also amended Figure 5 to include relative rainfall changes, and thereby help understand the choice of the boxes. The following text was added to the caption of Fig. 5: "…colours show relative (rather than absolute) rainfall differences, in multiples of the reference value".

*Page 3, lines 1-3: This is the main issue in the paper. You state that "a substantial part of today's Sahel moisture is sourced from the Mediterranean", and this has been shown to be one of the key areas in future GW scenarios (Park et al. 2016). Then you state that the flux from the Mediterranean is negligible compared to the tropical Atlantic. This should be substantiated. A comparison between tropical Atlantic and Mediterranean moisture sources should be shown, as well as a comparison between the effects on precipitation of the SST warming in both the basins.*

We did not want to imply that the moisture flux from the Mediterranean is negligible; only that the simulated increase in moisture flux from the Mediterranean is smaller than the increase in moisture flux from the Atlantic. Indeed, when looking at the absolute magnitude of the moisture flux (rather than the change), it becomes visible that while in the past there was even a larger moisture flux from the Mediterranean than from the Atlantic into the area under consideration, this changes in the future, when both regions contribute similarly to the moisture influx (figure below). This is consistent with our hypothesis of the establishment of a substantial monsoon circulation in the northern part of the Sahel, which thus becomes more connected with the Atlantic moisture source.

Nevertheless, we acknowledge the role that the Mediterranean moisture source plays in *setting the stage* for this monsoon expansion: As SSTs warm in the Mediterranean as well as the Atlantic, both regions can supply more moisture and, thereby, latent heat to the continent. This increased latent heating of the Sahelian troposphere can then trigger the proposed feedback mechanism by drawing in more moist air from the tropical North Atlantic. Such a generalized view – an initial moisture increase supplied from multiple sources, triggering an enhanced monsoon inflow from the Atlantic – is also consistent with the simulated changes in lower troposphere winds: we see an increase in (south-)westerly flow from the North Atlantic, but no increase in winds from the Mediterranean. Moreover, it is consistent with a mechanism already proposed by (Rowell, 2003) when discussing the impact of Mediterranean SSTs on Sahel rainfall.

We have amended the manuscript to account for this more general picture, and thank the reviewer for the useful remarks. With respect to the suggested "comparison between the effects on precipitation of the SST warming in both the basins": This would, in the strict sense, only be

possible by running independent model simulations where SST in either one of the basins is held fixed while the other basin is warming. However, we have included the below figure (absolute moisture fluxes; new Fig. 6 in the revised manuscript), and have amended Fig. 4, and believe that these already provide a good indication of the relative role of both basins.

[Figure]

**Figure above: As figure 4 (top panel) in the manuscript, but showing the absolute magnitude of moisture flux in the past (1850-1999; top) and the future (2070-2099; bottom).**

*Figure 2: Is the magnitude of drought computed as drought minus no-drought? So models reproducing drought should give negative values. I think it would be better to change the sign, also for coherence with the changes by the end of the 21st century. Moreover, in the text you state that Wet7 models are "better than average in reproducing drought magnitude", therefore I suggest to add the multimodel mean to the plot, to show this.*

In fact, in this figure, the magnitude of drought was computed as no-drought minus drought. We admit that this choice was somewhat counter-intuitive, and have changed it. We have also indicated in the figure the median deviation from the observed drought magnitude across the models; such that it becomes visible which models have a smaller-than-median deviation. The updated figure and caption are as follows (including new numbering since we have swapped figures 1 and 2):

[Figure]

**Figure 1.** Past and future Sahel summer rainfall in CMIP5 coupled climate models. Blue bars show the change in central and eastern Sahel average summer rainfall (0-30°E, 10-20°N, July–September) by the end of the century (2071–2095) under RCP8.5 compared to the 1901–1999 average, as fraction of that average. The seven models investigated in detail in this paper (Wet7 subset) are marked in bold. The solid orange line shows the difference (drought minus non–drought) between the 1970–1989 drought period and the rest of the observational period ("non–drought", 1901–1969 and 1990–2009), in mm/day. The dashed orange line shows the same difference divided by the standard deviation of the non–drought period, in units of standard deviations. Black horizontal lines show the respective observed values (CRU TS3.1, Harris et al. (2014)). The orange circle indicates the median deviation from the observed drought—non–drought difference across the model ensemble; i.e. models shown to the left of this point are closer than average to the observed value.

*Figure 6: How do you obtain precip-SST plots? Do they refer to 21st century only? Please clarify this.*

This refers to the whole period 1850-2100. Sorry for not being clear about this in the figure caption. We have amended the caption text.

*Figure 7: It would be very useful to add the SST time series to precipitation.*

Thanks for the suggestion. We have added the SST time series to this figure (Fig. 9 in the revised manuscript).

*Figure 8: I think showing again MIROC-ESM-CHEM is redundant, better to show the multimodel mean, alongside the conceptual scheme.*

> Averaging over multiple models would tend to iron out the individual models' internal variability and to obscure any potential correlation between a model's SST and rainfall on short time scales. Therefore we believe that the multi-model mean would not be representative of the dynamical processes we are interested in, and we prefer to compare a single model run with the conceptual scheme. We admit that the information in the left-hand panels of Figure 8 (new Figure 10) has been shown in previous figures, but argue that Figure 8 primarily serves the comparison between the model data and the theoretical concept, and for that purpose it is advantageous to combine both in one figure.

> That being said, we have revised Figure 8 (new Figure 10) to show more stylized versions of the MIROC-ESM-CHEM graphs, with the conceptual graphs overlaid on them. This should help to further draw attention to the qualitative comparison, and away from the specific model results which are already presented in previous figures.

Referee #2 (anonymous)

*1 - Authors have chosen the wettest models (wet7 subset of models from the CMIP5 ensemble) in the Sahelian region by the end of 21th century. Any other criterion could be chosen (e.g. the driest models). There is no explicit evidence in the paper why the wet7 subset is the one from which one expects better future rain predictions in Sahel. Models have biases, both in the average, standard deviation, extremes etc. as we compare model simulations with a reference observed period (like that shown in Fig. 1: from ~1900 to ~2000).*

> We do not aim to select the models from which we expect better future predictions. Rather, we single out those models which exhibit a pronounced increase in Sahel rainfall in the future (and in particular, in areas which today receive little rainfall), and investigate the potential reason why these models behave differently than the majority of other models, which only show weak trends. The fact that the Wet7 models perform better than average in reproducing the magnitude of the 70s/80s drought is an interesting additional finding, but was not a criterion in the selection of the models.

> We have amended and restructured the respective section of the manuscript and hope that the choice of models becomes clearer now. We have also swapped Figures 1 and 2 to align better with the discussion in the text.

*Moreover, authors say in pg. 2 lines 19-21: 'At the same time, we note that the Wet7 models perform better than average in reproducing the magnitude of the 1970–1989 drought period'. I should stress that, from Fig. 1 you cannot conclude the above sentence from information displayed on Fig. 1.*

It's true that this figure (previous figure 1/new figure 2) only highlights the performance of one model (MIROC-ESM-CHEM). That is why we cited both figures 1 and 2 in this sentence. The two figures together support this sentence.

*Therefore, a simple table of statistical biases should be included to clarify the performance of wet7 subset in the reference period as compared with other models.*

We believe that the new figure 1 (revised version of previous figure 2) serves the purpose of comparing all the models' performance for the historical drought period, and might be better accessible than a table. We note that a more comprehensive evaluation of model performance is not trivial (as the referee mentions, there are many different statistical properties that could be investigated, not to mention numerous dynamical features relevant to Sahel rainfall) and is not the focus of our present study. As mentioned in the response to referee #1 (and in the revised manuscript), previous studies have evaluated these models in more detail and may serve as a reference.

*2 - The non-linear response P-SST (on Fig. 8), which is the main paper's result is quite interesting but deserves much explanation. Only a single phrase at the end of Section 2 refers Fig. 8 and the non-linear relationship. Authors should clarify some points.*

We are sorry that the paper might have been too concise on this point. We have expanded the text at the end of section 2 as well as the corresponding figure caption (which is now Figure 10, due to the addition of two more figures).

*- Fig. 8 shows the non-linear relationship P-SST. It is evident that some smoothing and composite averaging is done to minimize noise. Please clarify that. Do P and SST are taken over some running averages? Do results change if the binning length is shortened? What is the delay in the proposed moisture-advection feedback?*

No running average is applied to P or SST before the binning is done. If the binning length is shortened, results do not change qualitatively, although the picture indeed becomes noisier (see example below) and each bin contains less data points on average.

[Figure]

**Figure above: As Figure 6 (top) in the paper but with bin width reduced by half, and showing results for all bins regardless of the number of data points that a bin contains (whereas in the manuscript only bars with at least 5 underlying data points are shown).**

The moisture-advection feedback arises from energy balance considerations (Levermann, Schewe, Petoukhov, & Held, 2009; Schewe, Levermann, & Cheng, 2012) and has no inherent time scale. Thus there is no delay: If in a given year conditions are favorable (e.g. high SST leading to high evaporation rate over ocean and high moisture supply to continent) then a continental monsoon can develop in that year, independent of previous or successive years.

If delay refers to the typical travel time of the monsoon circulation that carries the moist air and is fueled by latent heat release over the continent, then that time scale should be on the order of a few days to weeks, given typical near-surface wind velocities of a few meters per second.

*Minor points*

*4 - In pg. 1, line 13: Precise in the text the periods with episodes of heavy rainfall in the text.*

We have amended the sentence as follows: "…episodes of abundant rainfall such as in the 1930s and 50s, and even destructive rain and flood events such as in 2007 (Tschakert et al., 2010; Tarhule, 2005)."

*5 - Fig. 1 In the caption, the thick grey curves are quite indicative of the trend. However, the light grey lines for the two sets of models are totally overlapped becoming useless. It would be much clear to show the temporal curves of the interval range, i.e. the minimum and the maximum over each model set (the 7-model set and the 23-model set).*

We have changed the figure accordingly; now only the envelope and the mean of each model subset are shown.

*6 - pg. 2, lines 27-28 Authors say: 'This suggests that the rainfall increase is not simply a consequence of thermodynamic changes, but part of a shift in West African monsoon circulation dynamics' Justify the first sentence please.*

In a warming atmosphere, one could expect to see increasing rainfall simply due to the higher water-holding capacity of the air, according to the Clausius–Clapeyron relation. If this were the

only cause of the rainfall increase in the Sahel, we would not expect to see substantial changes in wind speed; moisture transport would only increase because of the higher moisture content of the monsoon winds. The fact that we observe increasing wind speeds towards the north and east of the present-day monsoon region indicates a spatial extension of the monsoon domain.

We have amended the sentence: "This suggests that the rainfall increase is not simply a consequence of thermodynamic changes (higher water–holding capacity of warmer air), but goes together with a shift in West African monsoon circulation dynamics."

*7 - Fig 2. Caption is 'Solid black line shows the difference in average summer precipitation (mm/day, averaged over 0-30_E, 10-20_N) between the 1970–1989 drought period and the rest of the observational period ("non–drought", 1901–1969 and 1990–2009).'Therefore we expect a negative anomaly: 'drought period' minus 'observed nondrought period'. Like in other parts of the paper, it is not understood what Is the subtrahend and the minuend of the subtraction.*

We apologize for the lack of clarity here. As mentioned above in response to referee #1, we have revised this figure and the corresponding caption. The caption now clearly states what is the subtrahend and the diminuend; and the choice of sign is more intuitive (negative values for drought).

*8 - Fig. 3. Caption: Change the word 'Difference' to the word 'Deviation from'. Difference between A and B is A minus B, so please clarify the caption.*

We have changed the caption as follows: "Change (future minus past) in average Sahel daily precipitation between the end of the 20th century (1970–1999) and the end of the 21st century (2070–2099)…".

*9 - Fig. 4 There is no grey color bar for the SST anomalies. At least indicate where is the zero value.*

We have included a greyscale color bar for the SST anomalies in Fig. 4.

Short comment #1 (P.-A. Monerie)

*Page 1 - There is a typo, line 22 "(Biasutti, 2013) - line 24, the reference should be placed after "Sahel rainfall". You can also cite Fontaine et al. (2011), among others Fontaine B, Roucou P, Monerie P-A (2011) Changes in the African monsoon region at medium-term time horizon using 12 AR4 coupled models under the A1b emissions scenario. Atmos Sci Lett 12:83–88. doi:10.1002/asl.321*

Thank you very much for this hint. We have included this reference, and corrected the placement of the citation.

*Page 2 - Line 18 "The positive trend.." is in fact obtained in at least 80 % of the CMIP5 simulations in Biasutti (2013),it is not only due to the wet7*

We agree that a majority of the models show a positive trend – as is also visible in our new Figure 1 (previously Figure 2). We refer here to the *ensemble mean* trend and the fact that only a few models exhibit a *substantial* positive trend. E.g., only five models (which all belong to the Wet7 subset) exhibit a rainfall increase larger than 50% by the end of the century (new Fig. 1).

We have revised the corresponding text as follows, to improve clarity: "Taken together, these seven models—hereafter referred to as the "Wet7" subset—can largely account for the positive rainfall trend that has been found in the CMIP5 ensemble as a whole (cf. Roehrig et al., 2013; Park et al., 2015): The Wet7 multi–model mean shows a doubling of average summer rainfall by 2100 (Fig. 2). In contrast, the mean over the 23 other models exhibits only a weak wetting trend of less than 20%; trends in the individual models are small and some models even show a drying trend."

*Line 19-21: The Wet7 is able to reproduce the 1970-1989 drought magnitude, but what is your conclusion? Do you think these models projections to be more reliable? Do there is a link between the projection and a models ability to reproduce the current climate?*

We do not think that a model's ability to reproduce past and current climate necessarily means that its projections will be reliable. Conversely, if a model does *not* reproduce past and current climate realistically, its projections may be treated with particular scrutiny. Thus, as mentioned in our response to Reviewer #1, the observation that the Wet7 models perform relatively well in reproducing the drought magnitude is not by itself a proof of the quality of their projections; but it is a motivation to take their projections into serious consideration and to further investigate why these projections differ from the rest of the ensemble.

We have amended the corresponding paragraph in the manuscript in order to be clearer about the rationale: "Although we focus here on the future projections, we also note that the Wet7 models perform better than average in reproducing the magnitude of the 1970–1989 drought period, the three MIROC models especially being very close to observed values (orange lines in Fig. 1, and inset in Fig. 2). This observation is consistent with a more comprehensive analysis of the CMIP5 models for the historical period (Biasutti, 2013), which found that past multi–decadal variability is underestimated by all except a few models, one MIROC model among them. It may serve as an additional motivation to further study the future projections by these models, which we do in the following. We point out, however, that there is much variation among the Wet7 models themselves in terms of past and projected rainfall changes, and the dynamical features discussed below may be more or less developed in different models. We use the wettest model, MIROC-ESM-CHEM, to illustrate our discussion, and show the other six models as evidence that our findings are not exclusive to just one model."

*Line 22: "The seasonal distribution.." It is also the case for the other models (not only with the Wet7). You do not comment the large spread obtained with Figure3.*

The seasonal distribution indeed shows a similar shape as in the Wet7 also in *some* of the other models – but not in all models: See the figure below. This figure also shows again how the Wet7

models (in particular the wettest 5 models) stick out from the ensemble simply in terms of the magnitude of the rainfall change.

In any case, our statement "The seasonal distribution of the rainfall change in the Wet7 shows a clear monsoonal shape" is valid. We mention this not so much as a distinction against the other models, but as an indication of the dynamical change that is behind the simulated rainfall change, namely an intensification and expansion of the West African monsoon, as discussed in the remainder of the paragraph.

We have amended the statement to account for the spread in Figure 3: "The seasonal distribution of the rainfall change in the Wet7 shows a clear monsoonal shape, despite considerable spread in its magnitude (Fig. 3)."

We have also slightly changed Figure 3: previously, it showed rainfall averaged over the rectangular area indicated in Figure 4, i.e. a slightly different area for each model. This was inconsistent with the main text, which we apologize for. Now, consistent with the text, Figure 3 shows rainfall averaged over the common region 10-20°N, 0-30°E as in Figure 1 and 2. This change does not affect the above statement or any other findings in the paper.

[Figure]

**Figure above: As figure 3 in the manuscript but based on monthly rainfall (not daily) and including all models; the Wet7 models are indicated by thick lines in bluish colors. Rainfall is averaged over 10-20°N, 0-30°E for all models.**

*Page3 - Line 1-3: Is it in contradictory with Park et al. (2016)?*

See our response to reviewer #1 (4[th] point). We have amended the manuscript (including additional figures) to account for the role of Mediterranean SSTs in supplying additional moisture and thus setting the stage for the dynamic response of the monsoon circulation. This proposed mechanism is in fact consistent with the importance of Mediterranean SSTs emphasized by Park et al. (2016).

*Line 4: Are you analysing the global SSTs? This sentence is not clear Did you found the same results focusing on the North Atlantic Ocean, or the Mediterranean Sea?*

> We are analyzing SSTs only in the tropical North Atlantic and (new) Mediterranean moisture source regions shown as boxes in Fig. 4 and 5; not global SSTs. We have indicated this in the main text and in the caption of Fig. 7. As we show in the revised manuscript, we find a similar rainfall-SST behavior for the tropical North Atlantic and the Mediterranean, but only the Atlantic moisture flux increase is linked to an increase in wind speed.

*Figure - Figure7: if we only consider the period with a contiuous increase in the GHGs concentration, (the RCP8.5 emission scenario starts in 2005-2006), is the precipitation increase so abrupt?*

> $CO_2$ concentrations have increased continuously throughout the simulation period. As new Fig. 9 (previously Fig. 7) shows, precipitation begins increasing substantially in the early 21[st] century. If we were to remove the years before 2005 from the analysis, we would still see the increase in precipitation, but we would lack the long period of relatively stable precipitation before the 21[st] century. It would then be hard to define what 'abrupt' means, as there would be no historical period for comparison. The abruptness, in the time domain, is seen in the long period of relatively stable precipitation followed by a steep increase.

Jacob Schewe (on behalf of all authors)

*Correspondence to:* Jacob Schewe (jacob.schewe@pik-potsdam.de)

**Abstract.** Projections of the response of Sahel rainfall to future global warming diverge significantly. Meanwhile, paleoclimatic records suggest that Sahel rainfall is capable of abrupt transitions in response to gradual forcing. Here we present climate modeling evidence for the possibility of an abrupt intensification of Sahel rainfall under future climate change. Analyzing 30 coupled global climate model simulations, we identify seven models where central Sahel rainfall increases by 40% to 300% over the 21[st] century, owing to a northward expansion of the West African monsoon domain. Rainfall in these models is non–linearly related to sea surface temperature (SST) in the tropical Atlantic and Mediterranean moisture source regions, intensifying abruptly beyond a certain SST warming level. We argue that this behaviour is consistent with a self–amplifying dynamic–thermodynamical feedback, implying that the gradual increase in oceanic moisture availability under warming could trigger a sudden intensification of monsoon rainfall far inland of today's core monsoon region.

**1  Introduction**

The Sahel is a wide semi–arid belt spanning the African continent south of the Sahara desert, and is home to a large population strongly reliant on agriculture (Sissoko et al., 2010). Its climate has been characterized by devastating droughts, such as in the 1970s and 80s (Folland et al., 1986; Zeng, 2003), alternating with episodes of abundant rainfall such as in the 1930s and 50s, and even destructive rain and flood events such as in 2007 (Tschakert et al., 2010; Tarhule, 2005). The 1970s/80s drought, which resulted in persistent food shortage and widespread famine (Nicholson, 2013), has been attributed to anthropogenic reflective aerosols as well as variations in Atlantic sea surface temperature (SST), which may have been partly human–induced and partly due to natural variability (Giannini et al., 2003; Biasutti and Giannini, 2006). Rainfall has partially recovered more recently (Lebel et al., 2009), a trend that has been attributed both to the direct radiative effect of anthropogenic greenhouse gases (Dong and Sutton, 2015) and to SST warming, especially in the Mediterranean (Park et al., 2016).

While coupled climate models generally capture the temporal pattern of the 1970s/80s drought (Biasutti and Giannini, 2006), most simulations from the recent Coupled Model Intercomparison Project phase 5 (CMIP5) underestimate its magnitude Biasutti (2013). At the same time, projections of future rainfall changes diverge substantially across the models (Power et al., 2012). The CMIP5 multi–model mean was shown to exhibit only a slight increase in overall Sahel rainfall (Fontaine et al., 2011; Biasutti, 2013), with a wetting trend over the central and eastern Sahel and drying over the westernmost part, under the

highest Representative Concentration Pathway (van Vuuren et al., 2011), RCP8.5. Individually, some of the models project a much stronger rainfall increase, while others even project an overall decrease.

This uncertainty in future projections raises questions about potential mechanisms of change that may be present in some models but not others, and that may be responsible for the large differences between models. In particular, paleoclimatic records suggest that Sahel rainfall is capable of abrupt transitions in response to gradual forcing (DeMenocal et al., 2000; McGee et al., 2013); and theoretical studies have demonstrated that such a non–linear response can in principle arise from internal monsoon dynamics (Levermann et al., 2009; Seshadri, 2016). In this study, we examine Sahel rainfall in state–of–the art climate model simulations and show that in those models that exhibit the strongest rainfall increase, this increase is non–linearly related to the SST warming in the tropical North Atlantic and Mediterranean moisture source regions. We argue that this behaviour is consistent with the theory and paleoclimatic evidence mentioned above. Considering that this non–linear rainfall response may be more pronounced in some models than in others may contribute to understanding the differences between the models' future projections.

**2    Methods and Results**

We investigated Sahel rainfall in 30 Coupled Model Intercomparison Project phase 5 (CMIP5) global climate models under RCP8.5 (see Appendix A). Three models (MIROC-ESM-CHEM, MIROC-ESM, BNU-ESM) project an increase of over 100% in average summer (July–September) rainfall across the central and eastern Sahel by the end of the 21$^{st}$ century (Fig. 1). Four other models (FGOALS-g2, MIROC5, CanESM2, NorESM1-M) project slightly smaller rainfall increases but with similar patterns as the three wettest models (a pronounced rainfall increase north and west of the present core monsoon region, see below). Taken together, these seven models—hereafter referred to as the "Wet7" subset—can largely account for the positive rainfall trend that has been found in the CMIP5 ensemble as a whole (cf. Roehrig et al., 2013; Park et al., 2015): The Wet7 multi–model mean shows a doubling of average summer rainfall by 2100 (Fig. 2). In contrast, the mean over the 23 other models exhibits only a weak wetting trend of less than 20%; trends in the individual models are small and some models even show a drying trend.

Although we focus here on the future projections, we also note that the Wet7 models perform better than average in re-producing the magnitude of the 1970–1989 drought period, the three MIROC models especially being very close to observed values (orange lines in Fig. 1, and inset in Fig. 2). This observation is consistent with a more comprehensive analysis of the CMIP5 models for the historical period (Biasutti, 2013), which found that past multi–decadal variability is underestimated by all except a few models, one MIROC model among them. It may serve as an additional motivation to further study the future projections by these models, which we do in the following. We point out, however, that there is much variation among the Wet7 models themselves in terms of past and projected rainfall changes, and the dynamical features discussed below may be more or less developed in different models. We use the wettest model, MIROC-ESM-CHEM, to illustrate our discussion, and show the other six models as evidence that our findings are not exclusive to just one model.

The seasonal distribution of the rainfall change in the Wet7 shows a clear monsoonal shape, despite considerable spread in its magnitude (Fig. 3). Generally, the rainfall increase occurs over a broad region between 10 and 20°N, i.e., extending into today's Sahara desert (Fig. 4). Conversely, rainfall decreases somewhat in the more humid regions around the Gulf of Guinea and the West coast. This pattern corresponds to an inland shift compared to the present–day rainfall regime. At the

5   same time, the near–surface, southwesterly winds intensify in the northern and eastern parts of the Sahel, near the positive rainfall anomaly, while they do not change much near the coast (Fig. 5). This suggests that the rainfall increase is not simply a consequence of thermodynamic changes (higher water–holding capacity of warmer air), but goes together with a shift in West African monsoon circulation dynamics.

Sahel rainfall has been linked to Atlantic as well as Mediterranean SSTs via evaporation rate and moisture supply (e.g.

10   Giannini et al., 2003; Rowell, 2003). In order to examine temporal patterns of rainfall and SST change more closely, we average each model's summer rainfall over a rectangular subregion of the Sahel (solid boxes in Fig. 4 and 5). The subregions are chosen to encompass an area where the rainfall increase is substantial in both absolute (Fig. 4) and relative terms (Fig. 5), and to be similar in size and location across the different models' grids (except for CanESM2 where the rainfall increase is located further east than in the other models). Thus, the subregions are generally located northward of the present–day core

15   monsoon regions, which also see rainfall increases but less pronounced in relative terms. Similarly, we identify for each model one region in the tropical North Atlantic ocean and one in the Mediterranean Sea (dashed boxes in Fig. 4) as the main sources of additional moisture influx into the Sahel, based on the lower–troposphere moisture flux changes (arrows in Fig. 4).

Moisture influx from both sources into North Africa is projected to increase in the Wet7 models (Fig. 4). However, moisture flux from the Atlantic into the Sahel subregion increases more strongly than from the Mediterranean by the end of the 21$^{st}$

20   century (Fig. 4 and 6). Moreover, only the Atlantic branch is accompanied by an increase in near–surface wind speed (Fig. 5). Thus, while the increased moisture import from the Mediterranean appears to be due to higher SST and evaporation alone, increased wind speed further amplifies moisture import from the Atlantic; a mechanism already proposed by Rowell (2003).

Sahel rainfall generally increases as the surface of the oceanic moisture source regions warms (Fig. 7 and 8). But this relation is not linear. Rainfall shows little response to SST changes within a range of approx. 1°C around the present–day value; but

25   when SST increases beyond this point, rainfall shifts abruptly to a stronger level, where it then keeps increasing as SST rises further. Given the convex shape of the temperature forcing over time, the abruptness of the rainfall response is expected to be less apparent in the time domain; nonetheless, rainfall appears relatively stable over the historical period, before it begins increasing strongly in the 21$^{st}$ century (Fig. 9).

Numerous paleoclimatic reconstructions reveal abrupt shifts in monsoon systems in Asia (Gupta et al., 2003; Wang et al.,

30   2008) and Africa (DeMenocal et al., 2000; McGee et al., 2013; Weldeab et al., 2007) before and throughout the Holocene. In those cases, external forcing through changes in solar insolation was much more gradual than that associated with modern anthropogenic climate change. A physical mechanism has been proposed to explain such abrupt shifts in large–scale monsoon rainfall in response to gradual forcing (Levermann et al., 2009, 2016): While the summer monsoon circulation is initiated by differential warming of land and ocean in spring, it is latent heat release from precipitation that maintains the land–sea

35   atmospheric temperature contrast throughout the summer and thus drives the monsoon winds into the continental interior. The

monsoon winds in turn supply the moisture necessary to maintain precipitation. Summer monsoon rainfall is thus powered by a positive feedback between moisture inflow and atmospheric heating. This positive moisture–advection feedback gives rise to a threshold behaviour with respect to external quantities that govern the energy budget of the monsoon; in particular, in this simplified theory, there is a minimum atmospheric humidity in the oceanic moisture source region below which such a monsoon circulation cannot be maintained (Schewe et al., 2012).

This framework has been used to explain abrupt variations in monsoon strength documented in Asian speleothem (Schewe et al., 2012) and pollen records (Herzschuh et al., 2014), but has not yet been applied to modern monsoon systems. We suggest that it is also useful for understanding the projected Sahel rainfall changes in the Wet7 models. Today, the West African monsoon is most active between the Gulf of Guinea coast and the southern edge of the Sahel (Nicholson, 2013). Rainfall declines towards the continental interior, and while central and eastern Sahel rainfall still exhibits a clear seasonality, it is relatively weak and erratic (compared to e.g. the Indian monsoon with its intense rainfall throughout much of the subcontinent). An increase in evaporation due to ocean warming in the tropical North Atlantic and the Mediterranean increases moisture availability. Once atmospheric humidity exceeds the monsoon threshold even in the more continental parts of the Sahel, the moisture–advection feedback can amplify the monsoon response by enhancing the westerly monsoon winds and thus the moisture influx from the North Atlantic. These inland regions thereby become increasingly connected with the oceanic moisture source, and benefit from further increases in oceanic evaporation.

This framework can explain the observed shape of the rainfall response in both the time and SST domains (Fig. 10): The functional form of the rainfall–SST relationship found in the Wet7 models (most prominently in MIROC–ESM–CHEM) resembles the concave form and threshold behaviour that arises from the above theory; given the convex form of mean SST forcing under global warming, the resulting pattern of rainfall over time is one where rainfall is relatively stable and low up to a certain point and then starts rising quasi–linearly.

**3    Discussion and Conclusions**

This explanation of an abrupt intensification of inland monsoon rainfall in the Sahel region is consistent with studies suggesting a substantially wetter Sahel, and Sahara, region in past climates compared to today (DeMenocal et al., 2000; Gasse, 2000). It is also consistent with theories linking rainfall changes in the Sahel to a combination of a local (through radiative forcing changes) and a remote (through tropical SST impacts on atmospheric stability) forcing mechanism (Giannini, 2010; Giannini et al., 2013; Seth et al., 2010). In a warming world, the remote mechanism would increase atmospheric stability especially in places with oceanic influence, and make it harder for convection to set in. Acting in the other direction, the local mechanism would directly warm the surface and decrease vertical stability over land. The mechanism we suggest here would act on top of these two mechanisms, and help explain the abruptness of the Sahel rainfall response to global warming. It would particularly affect the more continental parts of the region. We note that part of the increased moisture influx is through westerly winds near 10°N, a flow called the West African Westerly Jet (Pu and Cook, 2010, 2012). While its intraseasonal dynamics are somewhat distinct from the more southerly monsoon flow across the Gulf of Guinea, on a seasonal timescale both are driven by the

pressure—and thus, temperature—gradient between the eastern Atlantic and the Sahel, and would be subject to the dynamical feedback mechanism described above. Consideration of this mechanism may help to make sense of the diversity of model projections, and eventually establish a more consistent understanding of the Sahel's future climate in a warming world.

We also note that the marked increase in Sahel rainfall begins at remarkably similar levels of SST change across the Wet7 models: Mostly at around, or just below 1°C of SST warming (Fig. 7). In order to put these regional climatic changes in the context of global anthropogenic warming, we also show the projected Sahel rainfall changes over global mean temperature (GMT) change (Fig. 11). Given the different regional distribution of the warming signal in different models and the fact that GMT and SST do not necessarily co–vary on an annual time scale, there is not as clear an association of the rainfall change with GMT change as with SST change. However, it may be noted that in many of the models the "Paris range" of 1.5–2.0°C of global warming UNFCC (2015) presents an approximate dividing line between the historical Sahel climate regime and a substantially wetter future climate.

**Appendix A: Models and data**

We analyzed simulations from the BNU-ESM, CanESM2, FGOALS-g2, MIROC-ESM-CHEM, MIROC5, MIROC-ESM, NorESM1-M (Wet7 subset), ACCESS1-0, ACCESS1-3, CESM1-BGC, CESM1-CAM5, CMCC-CM, CMCC-CMS, CMCC-CESM, CNRM-CM5, EC-EARTH, FIO-ESM, GFDL-CM3, GFDL-ESM2M, GFDL-ESM2G, GISS-E2-H, GISS-E2-R, HadGEM2-ES, HadGEM2-CC, inmcm4, IPSL-CM5A-LR, IPSL-CM5A-MR, IPSL-CM5B-LR, MRI-CGCM3, and MPI-ESM-MR global climate models, driven by historical forcing and the RCP8.5 greenhouse gas concentration scenario (Taylor et al., 2012; Meinshausen et al., 2011). Simulation data was obtained from the CMIP5 archive at http://cmip-pcmdi.llnl.gov/cmip5/. Where several realizations of the same model simulation were available, we used the r1i1p1 configuration, since this one was available from all models. Near–surface wind data was not available for FGOALS-g2, therefore we show 850mb wind. CRU TS3.1 monthly precipitation data was obtained from http://badc.nerc.ac.uk.

*Author contributions.* J. Schewe and A. Levermann designed the research. J. Schewe carried out the analysis and wrote the paper, with contributions from A. Levermann.

*Competing interests.* The authors declare that they have no conflict of interest.

*Acknowledgements.* We acknowledge the World Climate Research Programme's Working Group on Coupled Modelling, which is responsible for CMIP, and we thank the climate modeling groups for producing and making available their model output. J.S. received funding through the Leibniz society's EXPACT project (SAW-2013-PIK-5).

[revised manuscript text omitted]

**Figure 1.** Past and future Sahel summer rainfall in CMIP5 coupled climate models. Blue bars show the change in central and eastern Sahel average summer rainfall (0-30°E, 10–20°N, July–September) by the end of the century (2071–2095) under RCP8.5 compared to the 1901–1999 average, as fraction of that average. The seven models investigated in detail in this paper (Wet7 subset) are marked in bold. The solid orange line shows the difference (drought minus non–drought) between the 1970–1989 drought period and the rest of the observational period ("non–drought", 1901–1969 and 1990–2009), in mm/day. The dashed orange line shows the same difference divided by the standard deviation of the non–drought period, in units of standard deviations. Black horizontal lines show the respective observed values (CRU TS3.1, Harris et al. (2014)). The orange circle indicates the median deviation from the observed drought—non–drought difference across the model ensemble; i.e. models shown to the left of this point are closer than average to the observed value.

[Figure]

**Figure 2.** Sahel summer rainfall in two model subsets and in observations. Shading shows the envelopes (minimum and maximum among models) of Sahel summer rainfall (0-30°E, 10–20°N, July–September) in the Wet7 subset (blue) and in 23 other CMIP5 models (grey) under historical forcing and the RCP8.5 greenhouse gas concentration scenario. Shown is the deviation from the 1900–1999 average, as fraction of that average. The thick lines indicate the averages of each set of models. Data between 1850–1860 and between 2095-2100 was unavailable for some of the models, therefore the grey line and shading only extend from 1860–2095. The inset shows the CRU TS3.1 observational data set covering 1901–2009 (red) and the corresponding portion of the MIROC–ESM–CHEM simulation (blue), in the same units as the other data but offset by 1.5 in the vertical for clarity.

[Figure]

**Figure 3.** Change (future minus past) in average Sahel ((0-30°E, 10–20°N) daily precipitation between the end of the 20th century (1970–1999) and the end of the 21st century (2070–2099), in the Wet7 models. All timeseries filtered with a 6–week running mean.

[Figure]

**Figure 4.** Simulated changes in Sahel summer climate under RCP8.5 in the Wet7 models. For each model the differences in July–September rainfall (colours), sea surface temperature (SST, greyscale; contour spacing is 0.5K), and moisture flux integrated vertically over the three bottom–most pressure levels (1000, 925, and 850 mb; arrows) are shown between the 20th century (1900–1999) and the end of the 21st century (2070–2099). Solid (dashed) boxes show the regions over which rainfall (SST) differences are averaged for Fig. 7 and the following figures. The color scale, vector scale, and coordinate labels of the top panel apply to all panels.

[Figure]

**Figure 5.** As Fig. 4, but colours show relative (rather than absolute) rainfall differences, in multiples of the reference value; and arrows show changes in near–surface winds (850mb winds for FGOALS-g2, where near–surface wind speed was not available).

[Figure]

**Figure 6.** As top panel in Fig. 4 but showing absolute moisture flux in the 20th century (1900–1999, top) and the end of the 21st century (2070–2099, bottom).

[Figure]

**Figure 7.** Median Sahel July–September rainfall for different intervals of tropical North Atlantic SST change (interval width $0.25\,^\circ C$), including all data between 1850 and 2100. Bars illustrate the deviation from the 1900–1999 rainfall average (horizontal black line), and SST change is also relative to the 1900–1999 average. Bars are only shown if at least 5 years fall into the respective temperature interval. Rainfall and SST are averaged over the corresponding boxes shown in Fig. 4 and 5.

[Figure]

**Figure 8.** As Fig. 7 but for Mediterranean SST change.

[Figure]

**Figure 9.** Mean Sahel July–September rainfall in 15-year intervals. Bars illustrate the deviation from the 1900–1999 rainfall average (horizontal black line), in mm/day. Blue line shows yearly July–September tropical North Atlantic SST, in °C (same axis).

[Figure]

[Figure]

[Figure]

**Figure 10.** Comparison of model simulations with the concept of a monsoon threshold. Data are from the MIROC-ESM-CHEM simulation, and are identical to the data shown in the top panels of Fig. 7 and 9; they are shown here without labels to emphasize the functional form. Orange lines (with blue axes) show illustrative functional forms that qualitatively match those of the simulation data, and are consistent with analytical results from a minimal monsoon model (Schewe et al., 2012): The moisture–advection feedback implies that no continental monsoon exists below a certain threshold (blue tick mark) in the energy budget—here controlled by sea surface temperature (SST)—, whereas above the threshold, monsoon intensity is a concave function of SST (top). In combination with a convex SST evolution (middle), this behaviour can give rise to the observed rainfall evolution over time, where rainfall is relatively stable and low up to a certain point and then starts rising quasi–linearly.

[Figure]

**Figure 11.** As Fig. 7, but with global mean temperature (GMT) change on the horizontal axis, instead of SST change. The blue shading marks the "Paris range", i.e. the global warming levels consistent with the United Nations' 2015 Paris Agreement (UNFCC, 2015).

---

## Referee Report (RR1)

**2nd Review of the paper "Non–linear intensification of Sahel rainfall as a dynamic response to future warming" by Jacob Schewe and Anders Levermann**

The authors carefully took into account and precisely responded to all my comments, and the manuscript results improved in comparison with the previous version. However, some major issues are still open, specifically concerning the robustness of the results and the final conclusions of the study. I find the results presented in this paper very interesting, with potential impact in the understanding and modelling of climate change in the Sahel. Therefore, I recommend that the authors fix these issues before the paper is published. See details below.

**Major comments**

**a) Robustness of the results.**

**1) Model selection.** The authors select the Wet7 models on the basis of the projected precipitation at the end of the 21st century. Three models are selected because of the magnitude of the response (>100%), four models are selected on the basis of the spatial pattern, even though some are "drier" than other models not selected. E.g. it can be noticed that CanESM2 and NorESM1-M are "drier" than HadGEM and ACCESS models (see Figure 1). I find the selection method not robust because no quantitative approach is presented to measure similarity in patterns. Moreover, the reader cannot verify whether not selected models show more or less similar patterns. I suggest to present a robust quantitative method to select the "wet" model subset.

**2) Domain selection.** In my previous review, I requested to describe the methodology to select the domain, but the authors' response is incomplete. Although the choice of the boxes appears quite obvious (especially for the oceanic domains), and small differences in size/position should not produce large differences, I think that the domain selection should not just be based on "substantial" differences. Does "substantial" mean that differences are statistically significant? Or above a threshold? This should be clarified, not to give the impression of a subjective choice. Moreover, presenting relative differences does not help much. Indeed, for some models the highest differences are outside the boxes, in deserted areas where even very small absolute differences mean large relative differences (see Figure 5).

**3) Rainfall-SST relationship.** When presenting the precipitation-GMT relationship (Figure 11), authors state that non-linearity is not reproduced as in the precipitation-SST plots (Figures 7 ad 8). Actually, I cannot see such a large difference between the two cases. Also in this case I recommend the statement to be supported by a quantitative demonstration. A simple functional shape for the precipitation-SST relationship could be hypothesised and tested. This method could also help to identify the specificity of the Atlantic and Mediterranean basins, in driving the increase in precipitation, compared to the global warming.

**4) Comparison with "neutral" and "dry" models.** In order to understand the specificity of the Wet7 subset, a comparison with the "neutral" and "dry" model subsets would be ideal. Indeed, the reader has no mean to understand whether the Wet7 models project such large anomalies because of their ability in simulating the precipitation-SST non-linearity, or

because they are just "warmer" than the others. In other word, does the precipitation-SST non-linearity hold also in "neutral/dry" models? Or are these model just not "warm" enough?

**b) Conclusions.** While the objective of the study is clearly stated in the Introduction, the conclusions (and the title as well) are quite misleading concerning the achievements of the study. The reader could have the impression that this study explain how precipitation in the Sahel will increase in the 21st century, but this is not the case. Indeed, on the one hand, the authors clearly state that they aim to identify the key mechanism leading some models to be wetter than others. On the other hand, they conclude that "this explanation of an abrupt intensification of inland monsoon rainfall in the Sahel region is consistent with studies suggesting a substantially wetter Sahel, and Sahara, region in past climates compared to today (DeMenocal et al., 2000; Gasse, 2000). It is also consistent with theories linking rainfall changes in the Sahel to a combination of a local (through radiative forcing changes) and a remote (through tropical SST impacts on atmospheric stability) forcing mechanism (Giannini, 2010; Giannini et al., 2013; Seth et al., 2010)"; and that "the mechanism we suggest here would act on top of these two mechanisms, and help explain the abruptness of the Sahel rainfall response to global warming. It would particularly affect the more continental parts of the region". They finally reconcile with the initial objective stating that "consideration of this mechanism may help to make sense of the diversity of model projections, and eventually establish a more consistent understanding of the Sahel's future climate in a warming world". I recommend the authors to make clear in the Conclusions that their results focus on the mechanisms underlying the projections of a "wet" model subset, not implying that these models are the good ones.

**Minor comments**

Page 2, line 18: "pronounced rainfall increase north and west", I see the rainfall increase north and east actually.

Figure 1: I don't see the discussion on the ability in modelling of drought crucial for the paper, which basically focuses on trends rather than multidecadal oscillations. Moreover, the discussion on the ability to reproduce multidecadal oscillations is rather concise and does not add much to the discussion. Even the ability of the Wet7 subset in simulating the big drought is questionable. Indeed, though Wet7 are above the average, FGOALS-g2 simulates almost no drought, and CanESM2, BNU-ESM and NorESM1-M are just slightly dry. I suggest to remove this figure, because not crucial for the model selection.

Figure 2: The MIROC5 ability in simulating the multidecadal variability is quite good in the second half of the century, and less good in the first half. Moreover, the assessment is only qualitative, not supported by any metric. Similarly to the comment above, I suggest to remove this argument to support the model selection.

Figures 4-6: Statistical significance of the differences should be assessed.

Figures 7, 8, 9 and 11: XY axes should share the same range, for better comparison.

Conclusions: The inclusion of the Mediterranean role in the discussion is helpful for understanding the main idea of the study in a broader context. I think that this could be emphasised in the conclusions.

Figure S2: is it the same as Figure 3?

Figure S3: in caption, reference should be to Figure 4 (not Figure 2).

Figure S4: it is redundant with Figure 9. I see more useful to show the evolution of rainfall along with the Mediterranean SST and the GMT (similarly to Figure 9).

---

## Editor Decision (ED1)

**ESD-2016-59: Editor Decision**

Vienna, March 31$^{st}$, 2017

Dear Authors,

Thank you very much for your thoughtful responses to reviewer and community remarks raised during the open discussion stage of the peer review process, and for having taken into consideration the editorial comments in the preliminary access review.

The discussion stage brought out a fertile and pertinent debate with respect and elevation among all intervening parties, which is very well appreciated. Insightful, constructive remarks have been formulated, and the authors aptly responded with thoughtful arguments and clear action towards further maturing the manuscript.

Overall, the manuscript is a very good work with relevance to Earth System Dynamics. Even so, it still holds potential to further excel through revision along the lines debated throughout the discussion stage. The authors are already undergoing significant and wise revision efforts, evident in their thorough final responses to the discussion stage.

All in all, by having carefully analysed the manuscript and the thorough discussion process, I warmly encourage the submission of a revised manuscript along the lines debated in the discussion stage and pledged in the final author responses.

If you have any further questions, please do not hesitate to contact us.

With very best wishes,

The handling editor,
Rui Perdigão

P.S.: Specific technical comment:

Regarding the new images, which are well appreciated, I would like to comment on the Figure on Page 3 of the final author comments. This is a relevant addition to the manuscript and overall well presented. However, the arrows are difficult to see in the darker boxes, and along the 10°N parallel where the flux is predominantly zonal. Moreover, where the caption refers to "the future", I would recommend "the simulated future" or equivalent expression deemed suitable to express that this is not necessarily the future per se, but rather a simulated one.

---

## Author Response (AR2)

Dear Dr. Perdigão,

thank you for taking the time to handle our manuscript! Please find below our responses to the comments by referee #1.

When considering the reviewer comments, we realized that the title of our manuscript may have been somewhat misleading: As it was submitted, the title might give the impression that the paper describes how Sahel rainfall *will* respond to future global warming. That is not the case, and we apologize for any misunderstanding this may have caused. We suggest amending the title as follows:

"Non–linear intensification of Sahel rainfall as a **possible** dynamic response to future warming"

This should make it clear that we do not provide a definite outlook on the future of Sahel rainfall, but rather demonstrate the *possibility* of a strong, non-linear intensification in state-of-the-art climate models. This is the idea with which we had written the paper, as also reflected in the abstract; we hope this may already resolve a number of issues, as we will detail now.

In response to the referee's comments under a):

1) Since the manuscript does not yet include an overview of the spatial patterns of rainfall change in all models, we suggest adding a set of figures as Supplementary Material. These figures show the patterns of rainfall change for all 30 models (see attachment). This would allow the reader to better assess the Wet7 models in the context of the other models, thereby complementing Figure 1.
Apart from this, we would like to point out that the focus of the paper is not to prove that a non-linear intensification is going to happen in the future, nor to make any kind of statistical statement such as: "Most models show…" or even "X out of Y models show…". The aim of the paper is to point out a possibility of future monsoon dynamics and to suggest a mechanism for that possibility.
To this end, it is not our goal to define an objective method for model selection. Our goal is to show that there are a significant number of models that exhibit a non-linear intensification of Sahel rainfall; and that these models are not simply outliers with particularly poor performance. The selection of the Wet7 models for further analysis has been based on a combination of different aspects: The magnitude of change in Sahel rainfall over the 21$^{st}$ century, and the location and spatial pattern of the rainfall changes. These aspects are difficult to combine into a single, objective measure, and we see no particular merit in doing so. Instead, we try to be as transparent as possible about our methods and the models that we selected, so that the reader can follow our line of analysis and, if need be, repeat the analysis with a different set of models. The supplementary figures mentioned above should help in this regard. We believe that any rule of selection would lead to a loss of information because we would have to either leave models out that show the phenomena or find a complicated selection rule that would be quite artificial.

We further note that the rainfall change shown in Figure 1 refers to the average over a common region (0-30°E, 10–20°N) for all 30 models. While the referee rightly points out that CanESM2 and NorESM1-M are "drier" than some other models in Figure 1, we nonetheless selected

CanESM2 because it exhibits a strong rainfall increase further east, as visible in Figure 4. Similarly, we selected NorESM1-M because, although somewhat weak in magnitude, its pattern of rainfall change is similar to that of the other Wet7 models, in the sense that it shows an increase particularly in the north-central Sahel region that is rather dry today. Nonetheless, one might argue to exclude NorESM1-M from the Wet7, or to include an eighth model; there is no obvious dividing line, but rather a gradient between the models as to how pronounced they show the effect that we want to demonstrate here. We note that we do not apply any statistics across models; each model is analyzed on its own, so if a model were to be removed from or added to the Wet7 subset, the results for the other models would remain unchanged.

We have rephrased the section on model selection as follows, including reference to the new Supplementary Figures (see also the revised manuscript, attached):

"Three models (MIROC-ESM-CHEM,MIROC-ESM, BNU-ESM) project an increase of over 100% in average summer (July–September) rainfall across the central and eastern Sahel by the end of the 21st century (Fig. 1). **Some other models (notably FGOALS-g2, MIROC5, CanESM2, NorESM1-M) project slightly smaller rainfall increases but with similar patterns as the three wettest models: a pronounced rainfall increase north and east of the present core monsoon region (Fig. 2; notice that while CanESM2 has only a moderate wetting trend over the central Sahel region analyzed in Fig. 1, it shows a strong rainfall increase in the eastern Sahel). This is in contrast to the majority of models which do not show a marked rainfall increase outside the present–day monsoon region (Supplementary Figures). We select the above–mentioned seven models— hereafter referred to as the "Wet7" subset—for further investigation, in order to elucidate possible mechanisms behind the strong rainfall increase in presently dry regions**."

Note the reordering of the main Figures: previous Fig. 4 is now Fig. 2.

2) We would argue in a similar way with regard to the domain selection: There is no obviously "correct" way of selecting the spatial domains; instead, we show the locations of the domains we chose in relation to both the relative and the absolute rainfall change, so that the reader can see what is captured by these particular domains. In fact, we have already harmonized the locations and sizes of the terrestrial domains to some extent; the existing differences across models are due to differences in model resolution and to differences in the actual spatial patterns in which the rainfall changes occur (for example, we have chosen a more eastern domain for CanESM2). Our experience with different choices for these domains confirm the referee's notion that "small differences in size/position should not produce large differences" in the results.

3) We believe this might be a misunderstanding caused by ambiguous language in our final paragraph. We do not mean to say that the non-linearity is more or less pronounced in the SST or GMT domain. Rather, we refer to the observation in Figure 7 that "the marked increase in Sahel rainfall begins at remarkably similar levels of SST change across the Wet7 models: Mostly

at around, or just below 1°C of SST warming". In the GMT domain, the temperature at which the marked rainfall increase begins is less similar across the models, for the reasons that we mention.

We have amended the text as follows, and hope to be clearer now:

"Given the different regional distribution of the warming signal in different models and the fact that GMT and SST do not necessarily co–vary on an annual time scale, there is not as clear an association of the rainfall change with GMT change as with SST change: **The GMT level at which Sahel rainfall begins to increase strongly is somewhat different across the models**. However, it may be noted that in many of the models the "Paris range" of 1.5–2.0°C of global warming (UNFCC 2015) presents an approximate dividing line between the historical Sahel climate regime and a substantially wetter future climate."

4) It would indeed be desirable to understand better the specific reasons for why a model does or does not exhibit a strong non-linear Sahel rainfall increase. However, we are afraid this is beyond the scope of the present paper. We hope that the paper may prompt other researchers – in particular those who are more familiar with individual models, e.g. the model developers – to investigate these processes in more detail, perhaps using designated model experiments.

b) The referee is absolutely right that our study focuses "on the mechanisms underlying the projections of a "wet" model subset". As mentioned above, we suggest changing the title in order to not create a false impression of the scope of the study. We have also added the following text in the Conclusions section:

"The mechanism we suggest here would act on top of these two mechanisms, and help explain the abruptness of the Sahel rainfall response to global warming **seen in some models**."

"**We have found a strong, non-linear Sahel rainfall increase only in a minority of the CMIP5 models, and we do not mean to imply that these particular model simulations are more realistic than others.**"

Minor comments:

Page 2, line 18: "pronounced rainfall increase north and west", I see the rainfall increase north and east actually.

The referee is right, we have corrected this mistake.

Figure 1: I don't see the discussion on the ability in modelling of drought crucial for the paper, which basically focuses on trends rather than multidecadal oscillations. Moreover, the discussion on the ability to reproduce multidecadal oscillations is rather concise and does not add much to the discussion. Even the ability of the Wet7 subset in simulating the big drought is questionable. Indeed, though Wet7 are

above the average, FGOALS-g2 simulates almost no drought, and CanESM2, BNU-ESM and NorESM1-M are just slightly dry. I suggest to remove this figure, because not crucial for the model selection.

> We agree that the drought is not the focus of our paper. However, we believe that Figure 1 provides useful context for the entire model ensemble, and the relatively good performance of the Wet7 for the drought magnitude underlines the fact that these models are worth studying for this particular region, and are not some outliers which should not be trusted at all. We don't mean to suggest that the Wet7 are the "best" models. As stated in the manuscript, the analysis of the drought magnitude merely "may serve as an additional motivation to further study the future projections by these models, which we do in the following. We point out, however, that there is much variation among theWet7 models themselves in terms of past and projected rainfall changes".

Figure 2: The MIROC5 ability in simulating the multidecadal variability is quite good in the second half of the century, and less good in the first half. Moreover, the assessment is only qualitative, not supported by any metric. Similarly to the comment above, I suggest to remove this argument to support the model selection.

> Along the same lines as above, we would prefer to keep this part of the argument, because it contributes to elucidating the performance of the Wet7 models in the context of the full model ensemble. We would also not remove Figure 2 because it provides an overview of the model ensemble's future rainfall projections and thereby provides the reader with important contextual information. Previous reviewers also found this figure useful. We have rephrased the discussion of model selection, as mentioned above, and hope that it will become clear now that model selection is not based on the drought performance.

Figures 4-6: Statistical significance of the differences should be assessed.

> We believe that showing both absolute and relative rainfall differences gives an idea about the significance of the differences compared to the overall magnitude, and is indeed more meaningful than some statistical measure of significance.

Figures 7, 8, 9 and 11: XY axes should share the same range, for better comparison.

> Because the focus here is on the shape of the rainfall change over temperature over time, rather than on a comparison of the magnitude of change across models, we would prefer to keep the axes ranges as is, in order to provide good readability also for the lower-magnitude changes.

Conclusions: The inclusion of the Mediterranean role in the discussion is helpful for understanding the main idea of the study in a broader context. I think that this could be emphasised in the conclusions.

> Thanks for this suggestion, which we have taken up in the revised text:

**"Our analysis also provides further evidence for the role of the Mediterranean Sea as a contributor to enhanced Sahelian moisture availability, which then further amplifies inflow from the North Atlantic through latent heating over the Sahel."**

The referee also raises some questions referring to Figures S2, S3, and S4. We believe he is referring to an older manuscript version here, which was accompanied by a Supplement. In the last revision, we moved the Supplementary figures into the main manuscript, and there is no Supplement now.

Jacob Schewe, for the authors

01 June 2017

**Non–linear intensification of Sahel rainfall as a possible dynamic response to future warming**

Jacob Schewe[1] and Anders Levermann[1,2,3]

[1]Potsdam Institute for Climate Impact Research, Potsdam, Germany
[2]Institute of Physics, Potsdam University, Potsdam, Germany
[3]Lamont-Doherty Earth Observatory, Columbia University, New York, USA

*Correspondence to:* Jacob Schewe (jacob.schewe@pik-potsdam.de)

**Abstract.** Projections of the response of Sahel rainfall to future global warming diverge significantly. Meanwhile, paleoclimatic records suggest that Sahel rainfall is capable of abrupt transitions in response to gradual forcing. Here we present climate modeling evidence for the possibility of an abrupt intensification of Sahel rainfall under future climate change. Analyzing 30 coupled global climate model simulations, we identify seven models where central Sahel rainfall increases by 40% to 300% over the 21st century, owing to a northward expansion of the West African monsoon domain. Rainfall in these models is non–linearly related to sea surface temperature (SST) in the tropical Atlantic and Mediterranean moisture source regions, intensifying abruptly beyond a certain SST warming level. We argue that this behaviour is consistent with a self–amplifying dynamic–thermodynamical feedback, implying that the gradual increase in oceanic moisture availability under warming could trigger a sudden intensification of monsoon rainfall far inland of today's core monsoon region.

**1 Introduction**

The Sahel is a wide semi–arid belt spanning the African continent south of the Sahara desert, and is home to a large population strongly reliant on agriculture (Sissoko et al., 2010). Its climate has been characterized by devastating droughts, such as in the 1970s and 80s (Folland et al., 1986; Zeng, 2003), alternating with episodes of abundant rainfall such as in the 1930s and 50s, and even destructive rain and flood events such as in 2007 (Tschakert et al., 2010; Tarhule, 2005). The 1970s/80s drought, which resulted in persistent food shortage and widespread famine (Nicholson, 2013), has been attributed to anthropogenic reflective aerosols as well as variations in Atlantic sea surface temperature (SST), which may have been partly human–induced and partly due to natural variability (Giannini et al., 2003; Biasutti and Giannini, 2006). Rainfall has partially recovered more recently (Lebel et al., 2009), a trend that has been attributed both to the direct radiative effect of anthropogenic greenhouse gases (Dong and Sutton, 2015) and to SST warming, especially in the Mediterranean (Park et al., 2016).

While coupled climate models generally capture the temporal pattern of the 1970s/80s drought (Biasutti and Giannini, 2006), most simulations from the recent Coupled Model Intercomparison Project phase 5 (CMIP5) underestimate its magnitude Biasutti (2013). At the same time, projections of future rainfall changes diverge substantially across the models (Power et al., 2012). The CMIP5 multi–model mean was shown to exhibit only a slight increase in overall Sahel rainfall (Fontaine et al., 2011; Biasutti, 2013), with a wetting trend over the central and eastern Sahel and drying over the westernmost part, under the

highest Representative Concentration Pathway (van Vuuren et al., 2011), RCP8.5. Individually, some of the models project a much stronger rainfall increase, while others even project an overall decrease.

This uncertainty in future projections raises questions about potential mechanisms of change that may be present in some models but not others, and that may be responsible for the large differences between models. In particular, paleoclimatic records suggest that Sahel rainfall is capable of abrupt transitions in response to gradual forcing (DeMenocal et al., 2000; McGee et al., 2013); and theoretical studies have demonstrated that such a non–linear response can in principle arise from internal monsoon dynamics (Levermann et al., 2009; Seshadri, 2016). In this study, we examine Sahel rainfall in state–of–the art climate model simulations and show that in those models that exhibit the strongest rainfall increase, this increase is non–linearly related to the SST warming in the tropical North Atlantic and Mediterranean moisture source regions. We argue that this behaviour is consistent with the theory and paleoclimatic evidence mentioned above. Considering that this non–linear rainfall response may be more pronounced in some models than in others may contribute to understanding the differences between the models' future projections.

**2  Methods and Results**

We investigated Sahel rainfall in 30 Coupled Model Intercomparison Project phase 5 (CMIP5) global climate models under RCP8.5 (see Appendix A). Three models (MIROC-ESM-CHEM, MIROC-ESM, BNU-ESM) project an increase of over 100% in average summer (July–September) rainfall across the central and eastern Sahel by the end of the 21$^{st}$ century (Fig. 1). Some other models (notably FGOALS-g2, MIROC5, CanESM2, NorESM1-M) project slightly smaller rainfall increases but with similar patterns as the three wettest models: a pronounced rainfall increase north and east of the present core monsoon region (Fig. 2; notice that while CanESM2 has only a moderate wetting trend over the central Sahel region analyzed in Fig. 1, it shows a strong rainfall increase in the eastern Sahel). This is in contrast to the majority of models which do not show a marked rainfall increase outside the present–day monsoon region (Supplementary Figures). We select the above–mentioned seven models— hereafter referred to as the "Wet7" subset—for further investigation, in order to elucidate possible mechanisms behind the strong rainfall increase in presently dry regions. We note that the Wet7 can largely account for the positive rainfall trend that has been found in the CMIP5 ensemble as a whole (cf. Roehrig et al., 2013; Park et al., 2015): The Wet7 multi–model mean shows a doubling of average summer rainfall by 2100 (Fig. 3). In contrast, the mean over the 23 other models exhibits only a weak wetting trend of less than 20%; trends in the individual models are small and some models even show a drying trend.

Although we focus here on the future projections, we also note that the Wet7 models perform better than average in re-producing the magnitude of the 1970–1989 drought period, the three MIROC models especially being very close to observed values (orange lines in Fig. 1, and inset in Fig. 3). This observation is consistent with a more comprehensive analysis of the CMIP5 models for the historical period (Biasutti, 2013), which found that past multi–decadal variability is underestimated by all except a few models, one MIROC model among them. It may serve as an additional motivation to further study the future projections by these models, which we do in the following. We point out, however, that there is much variation among the Wet7 models themselves in terms of past and projected rainfall changes, and the dynamical features discussed below may be more

or less developed in different models. We use the wettest model, MIROC-ESM-CHEM, to illustrate our discussion, and show the other six models as evidence that our findings are not exclusive to just one model.

The seasonal distribution of the rainfall change in the Wet7 shows a clear monsoonal shape, despite considerable spread in its magnitude (Fig. 3). Generally, the rainfall increase occurs over a broad region between 10 and 20°N, i.e., extending into today's Sahara desert (Fig. 4). Conversely, rainfall decreases somewhat in the more humid regions around the Gulf of Guinea and the West coast. This pattern corresponds to an inland shift compared to the present–day rainfall regime. At the same time, the near–surface, southwesterly winds intensify in the northern and eastern parts of the Sahel, near the positive rainfall anomaly, while they do not change much near the coast (Fig. 5). This suggests that the rainfall increase is not simply a consequence of thermodynamic changes (higher water–holding capacity of warmer air), but goes together with a shift in West African monsoon circulation dynamics.

Sahel rainfall has been linked to Atlantic as well as Mediterranean SSTs via evaporation rate and moisture supply (e.g. Giannini et al., 2003; Rowell, 2003). In order to examine temporal patterns of rainfall and SST change more closely, we average each model's summer rainfall over a rectangular subregion of the Sahel (solid boxes in Fig. 4 and 5). The subregions are chosen to encompass an area where the rainfall increase is substantial in both absolute (Fig. 4) and relative terms (Fig. 5), and to be similar in size and location across the different models' grids (except for CanESM2 where the rainfall increase is located further east than in the other models). Thus, the subregions are generally located northward of the present–day core monsoon regions, which also see rainfall increases but less pronounced in relative terms. Similarly, we identify for each model one region in the tropical North Atlantic ocean and one in the Mediterranean Sea (dashed boxes in Fig. 4) as the main sources of additional moisture influx into the Sahel, based on the lower–troposphere moisture flux changes (arrows in Fig. 4).

Moisture influx from both sources into North Africa is projected to increase in the Wet7 models (Fig. 2). However, moisture flux from the Atlantic into the Sahel subregion increases more strongly than from the Mediterranean by the end of the 21$^{st}$ century (Fig. 2 and 6). Moreover, only the Atlantic branch is accompanied by an increase in near–surface wind speed (Fig. 5). Thus, while the increased moisture import from the Mediterranean appears to be due to higher SST and evaporation alone, increased wind speed further amplifies moisture import from the Atlantic; a mechanism already proposed by Rowell (2003).

Sahel rainfall generally increases as the surface of the oceanic moisture source regions warms (Fig. 7 and 8). But this relation is not linear. Rainfall shows little response to SST changes within a range of approx. 1°C around the present–day value; but when SST increases beyond this point, rainfall shifts abruptly to a stronger level, where it then keeps increasing as SST rises further. Given the convex shape of the temperature forcing over time, the abruptness of the rainfall response is expected to be less apparent in the time domain; nonetheless, rainfall appears relatively stable over the historical period, before it begins increasing strongly in the 21$^{st}$ century (Fig. 9).

Numerous paleoclimatic reconstructions reveal abrupt shifts in monsoon systems in Asia (Gupta et al., 2003; Wang et al., 2008) and Africa (DeMenocal et al., 2000; McGee et al., 2013; Weldeab et al., 2007) before and throughout the Holocene. In those cases, external forcing through changes in solar insolation was much more gradual than that associated with modern anthropogenic climate change. A physical mechanism has been proposed to explain such abrupt shifts in large–scale monsoon rainfall in response to gradual forcing (Levermann et al., 2009, 2016): While the summer monsoon circulation is initiated

by differential warming of land and ocean in spring, it is latent heat release from precipitation that maintains the land–sea atmospheric temperature contrast throughout the summer and thus drives the monsoon winds into the continental interior. The monsoon winds in turn supply the moisture necessary to maintain precipitation. Summer monsoon rainfall is thus powered by a positive feedback between moisture inflow and atmospheric heating. This positive moisture–advection feedback gives rise to a threshold behaviour with respect to external quantities that govern the energy budget of the monsoon; in particular, in this simplified theory, there is a minimum atmospheric humidity in the oceanic moisture source region below which such a monsoon circulation cannot be maintained (Schewe et al., 2012).

This framework has been used to explain abrupt variations in monsoon strength documented in Asian speleothem (Schewe et al., 2012) and pollen records (Herzschuh et al., 2014), but has not yet been applied to modern monsoon systems. We suggest that it is also useful for understanding the projected Sahel rainfall changes in the Wet7 models. Today, the West African monsoon is most active between the Gulf of Guinea coast and the southern edge of the Sahel (Nicholson, 2013). Rainfall declines towards the continental interior, and while central and eastern Sahel rainfall still exhibits a clear seasonality, it is relatively weak and erratic (compared to e.g. the Indian monsoon with its intense rainfall throughout much of the subcontinent). An increase in evaporation due to ocean warming in the tropical North Atlantic and the Mediterranean increases moisture availability. Once atmospheric humidity exceeds the monsoon threshold even in the more continental parts of the Sahel, the moisture–advection feedback can amplify the monsoon response by enhancing the westerly monsoon winds and thus the moisture influx from the North Atlantic. These inland regions thereby become increasingly connected with the oceanic moisture source, and benefit from further increases in oceanic evaporation.

This framework can explain the observed shape of the rainfall response in both the time and SST domains (Fig. 10): The functional form of the rainfall–SST relationship found in the Wet7 models (most prominently in MIROC–ESM–CHEM) resembles the concave form and threshold behaviour that arises from the above theory; given the convex form of mean SST forcing under global warming, the resulting pattern of rainfall over time is one where rainfall is relatively stable and low up to a certain point and then starts rising quasi–linearly.

**3 Discussion and Conclusions**

This explanation of an abrupt intensification of inland monsoon rainfall in the Sahel region is consistent with studies suggesting a substantially wetter Sahel, and Sahara, region in past climates compared to today (DeMenocal et al., 2000; Gasse, 2000). It is also consistent with theories linking rainfall changes in the Sahel to a combination of a local (through radiative forcing changes) and a remote (through tropical SST impacts on atmospheric stability) forcing mechanism (Giannini, 2010; Giannini et al., 2013; Seth et al., 2010). In a warming world, the remote mechanism would increase atmospheric stability especially in places with oceanic influence, and make it harder for convection to set in. Acting in the other direction, the local mechanism would directly warm the surface and decrease vertical stability over land. The mechanism we suggest here would act on top of these two mechanisms, and help explain the abruptness of the Sahel rainfall response to global warming seen in some models. It would particularly affect the more continental parts of the region. We note that part of the increased moisture influx is through

westerly winds near 10°N, a flow called the West African Westerly Jet (Pu and Cook, 2010, 2012). While its intraseasonal dynamics are somewhat distinct from the more southerly monsoon flow across the Gulf of Guinea, on a seasonal timescale both are driven by the pressure—and thus, temperature—gradient between the eastern Atlantic and the Sahel, and would be subject to the dynamical feedback mechanism described above. Our analysis also provides further evidence for the role of the Mediterranean Sea as a contributor to enhanced Sahelian moisture availability, which then further amplifies inflow from the North Atlantic through latent heating over the Sahel.

We have found a strong, non–linear Sahel rainfall increase only in a minority of the CMIP5 models, and we do not mean to imply that these particular model simulations are more realistic than others. Consideration of the mechanism demonstrated here may, however, help to make sense of the diversity of model projections, and eventually establish a more consistent understanding of the Sahel's future climate in a warming world. We also note that the marked increase in Sahel rainfall begins at remarkably similar levels of SST change across the Wet7 models: Mostly at around, or just below 1°C of SST warming (Fig. 7). In order to put these regional climatic changes in the context of global anthropogenic warming, we also show the projected Sahel rainfall changes over global mean temperature (GMT) change (Fig. 11). Given the different regional distribution of the warming signal in different models and the fact that GMT and SST do not necessarily co–vary on an annual time scale, there is not as clear an association of the rainfall change with GMT change as with SST change: The GMT level at which Sahel rainfall begins to increase strongly is somewhat different across the models. However, it may be noted that in many of the models the "Paris range" of 1.5–2.0°C of global warming (UNFCC, 2015) presents an approximate dividing line between the historical Sahel climate regime and a substantially wetter future climate.

**Appendix A: Models and data**

We analyzed simulations from the BNU-ESM, CanESM2, FGOALS-g2, MIROC-ESM-CHEM, MIROC5, MIROC-ESM, NorESM1-M (Wet7 subset), ACCESS1-0, ACCESS1-3, CESM1-BGC, CESM1-CAM5, CMCC-CM, CMCC-CMS, CMCC-CESM, CNRM-CM5, EC-EARTH, FIO-ESM, GFDL-CM3, GFDL-ESM2M, GFDL-ESM2G, GISS-E2-H, GISS-E2-R, HadGEM2-ES, HadGEM2-CC, inmcm4, IPSL-CM5A-LR, IPSL-CM5A-MR, IPSL-CM5B-LR, MRI-CGCM3, and MPI-ESM-MR global climate models, driven by historical forcing and the RCP8.5 greenhouse gas concentration scenario (Taylor et al., 2012; Meinshausen et al., 2011). Simulation data was obtained from the CMIP5 archive at http://cmip-pcmdi.llnl.gov/cmip5/. Where several realizations of the same model simulation were available, we used the r1i1p1 configuration, since this one was available from all models. Near–surface wind data was not available for FGOALS-g2, therefore we show 850mb wind. CRU TS3.1 monthly precipitation data was obtained from http://badc.nerc.ac.uk.

*Author contributions.* J. Schewe and A. Levermann designed the research. J. Schewe carried out the analysis and wrote the paper, with contributions from A. Levermann.

*Competing interests.* The authors declare that they have no conflict of interest.

*Acknowledgements.* We acknowledge the World Climate Research Programme's Working Group on Coupled Modelling, which is responsible for CMIP, and we thank the climate modeling groups for producing and making available their model output. J.S. received funding through the Leibniz society's EXPACT project (SAW-2013-PIK-5).

[revised manuscript text omitted]

**Figure 1.** Past and future Sahel summer rainfall in CMIP5 coupled climate models. Blue bars show the change in central and eastern Sahel average summer rainfall (0-30°E, 10–20°N, July–September) by the end of the century (2071–2095) under RCP8.5 compared to the 1901–1999 average, as fraction of that average. The seven models investigated in detail in this paper (Wet7 subset) are marked in bold. Filled triangles show the difference (drought minus non–drought) between the 1970–1989 drought period and the rest of the observational period ("non–drought", 1901–1969 and 1990–2009), in mm/day. Open triangles show the same difference divided by the standard deviation of the non–drought period, in units of standard deviations. The solid and dashed horizontal lines show the respective observed values (CRU TS3.1, Harris et al. (2014)). The orange circle indicates the median deviation from the observed drought—non–drought difference across the model ensemble; i.e. models shown to the left of this point are closer than average to the observed value.

[Figure]

**Figure 2.** Simulated changes in Sahel summer climate under RCP8.5 in the Wet7 models. For each model the differences in July–September rainfall (colours), sea surface temperature (SST, greyscale; contour spacing is 0.5K), and moisture flux integrated vertically over the three bottom–most pressure levels (1000, 925, and 850 mb; arrows) are shown between the 20th century (1900–1999) and the end of the 21st century (2070–2099). Solid (dashed) boxes show the regions over which rainfall (SST) differences are averaged for Fig. 7 and the following figures. The color scale, vector scale, and coordinate labels of the top panel apply to all panels.

[Figure]

**Figure 3.** Sahel summer rainfall in two model subsets and in observations. Shading shows the envelopes (minimum and maximum among models) of Sahel summer rainfall (0-30°E, 10–20°N, July–September) in the Wet7 subset (blue) and in 23 other CMIP5 models (grey) under historical forcing and the RCP8.5 greenhouse gas concentration scenario. Shown is the deviation from the 1900–1999 average, as fraction of that average. The thick lines indicate the averages of each set of models. Data between 1850–1860 and between 2095-2100 was unavailable for some of the models, therefore the grey line and shading only extend from 1860–2095. The inset shows the CRU TS3.1 observational data set covering 1901–2009 (red) and the corresponding portion of the MIROC–ESM–CHEM simulation (blue), in the same units as the other data but offset by 1.5 in the vertical for clarity.

[Figure]

**Figure 4.** Change (future minus past) in average Sahel (0-30°E, 10–20°N) daily precipitation between the end of the 20[th] century (1970–1999) and the end of the 21[st] century (2070–2099), in the Wet7 models. All timeseries filtered with a 6–week running mean.

[Figure]

**Figure 5.** As Fig. 2, but colours show relative (rather than absolute) rainfall differences, in multiples of the reference value; and arrows show changes in near–surface winds (850mb winds for FGOALS-g2, where near–surface wind speed was not available).

[Figure]

**Figure 6.** As top panel in Fig. 2 but showing absolute moisture flux in the 20th century (1900–1999, top) and the end of the 21st century (2070–2099, bottom).

[Figure]

**Figure 7.** Median Sahel July–September rainfall for different intervals of tropical North Atlantic SST change (interval width $0.25°C$), including all data between 1850 and 2100. Bars illustrate the deviation from the 1900–1999 rainfall average (horizontal black line), and SST change is also relative to the 1900–1999 average. Bars are only shown if at least 5 years fall into the respective temperature interval. Rainfall and SST are averaged over the corresponding boxes shown in Fig. 2 and 5.

[Figure]

**Figure 8.** As Fig. 7 but for Mediterranean SST change.

[Figure]

**Figure 9.** Mean Sahel July–September rainfall in 15-year intervals. Bars illustrate the deviation from the 1900–1999 rainfall average (horizontal black line), in mm/day. Blue line shows yearly July–September tropical North Atlantic SST, in °C (same axis).

[Figure]

[Figure]

[Figure]

**Figure 10.** Comparison of model simulations with the concept of a monsoon threshold. Data are from the MIROC-ESM-CHEM simulation, and are identical to the data shown in the top panels of Fig. 7 and 9; they are shown here without labels to emphasize the functional form. Orange lines (with blue axes) show illustrative functional forms that qualitatively match those of the simulation data, and are consistent with analytical results from a minimal monsoon model (Schewe et al., 2012): The moisture–advection feedback implies that no continental monsoon exists below a certain threshold (blue tick mark) in the energy budget—here controlled by sea surface temperature (SST)—, whereas above the threshold, monsoon intensity is a concave function of SST (top). In combination with a convex SST evolution (middle), this behaviour can give rise to the observed rainfall evolution over time, where rainfall is relatively stable and low up to a certain point and then starts rising quasi–linearly.

[Figure]

**Figure 11.** As Fig. 7, but with global mean temperature (GMT) change on the horizontal axis, instead of SST change. The blue shading marks the "Paris range", i.e. the global warming levels consistent with the United Nations' 2015 Paris Agreement (UNFCC, 2015).